**communications** engineering

# Transmissive metasurface with 3.5-μm-thick liquid crystals for subterahertz-wave dynamic beamforming
Daisuke Kitayama [1] ✉, Hibiki Kagami [1], Adam Pander[1], Yuto Hama [2] & Hiroyuki Takahashi[1]

Propagation control is essential for the practical use of subterahertz waves. Liquid crystal (LC) metasurfaces, which can be easily fabricated using display manufacturing technologies, can achieve the large aperture required for controlling propagation channels. Here we propose a dual-linear polarization unit cell that incorporates an LC layer with a thickness that can be determined independently of cell scaling, and that can be designed over a broad frequency range from microwave to subterahertz. Our prototype transmissive metasurface includes a 3.5-μm-thick LC layer and comprises 47,524 cells with a cell size below λ/8, and exhibits an insertion loss of 2.5 dB with a 3-dB bandwidth of 10%. We experimentally demonstrate two-dimensional beam steering up to 30 degrees and variable focusing through amplitude modulation of the aperture in the 115-GHz band. We anticipate that the development of metasurfaces with display-grade LC thickness will promote the industrial use of subterahertz bands in next-generation mobile communications.

Wireless technology is employed in many contexts and has facilitated substantial enhancements to the convenience and quality of life. However, the currently used frequency resource, which must be shared by various applications, is approaching its capacity limit, making it imperative to develop new frequency bands. The potential use of the subterahertz band in next-generation mobile communication systems is currently under discussion among industry, academia, and government[1,2]. Subterahertz waves can efficiently propagate only in line-of-sight; thus, the establishment of a means to control the propagation paths of these waves in indoor-to-outdoor and non-line-of-sight (NLoS) scenarios will be essential[3]. imensional

Metasurfaces are two-dimensional arrays consisting of subwavelength-sized elements with designable scattering characteristics. They can be installed on building windows or walls to provide a designed wireless environment[4–6]. Moreover, dynamic metasurfaces, which integrate active components or functional materials with metasurfaces, are expected to be a crucial technology for controlling the propagation of transmitted and reflected subterahertz waves. In particular, dynamic metasurfaces are referred to as reconfigurable intelligent surfaces (RIS) in the field of mobile communications[7,8]. A variety of ways of making metasurfaces electrically controllable have been studied, including the use of semiconductor-based devices[9–12], micro-electro-mechanical systems[13,14], vanadium dioxide[15–17], graphene[18–20], liquid crystals (LC)[21–30], and other materials. To reduce the total propagation loss of the path through the square metasurface to a level commensurate with the free space propagation loss of the total propagation path length, the side length of an ideally controlled metasurface should be greater than or equal to the radius of the 1st Fresnel zone (Supplementary note 1). For instance, in order to propagate 100-GHz waves a distance of 100 m, it would be necessary to fabricate a metasurface approximately 200 mm on a side. Here, LC is the most promising material for creating such a large dynamic metasurface for RISs, given the availability of mass-production technology for devices with much larger areas, such as LC displays (LCDs). LC molecular relative permittivity varies between $\varepsilon r_\parallel$ and $\varepsilon r_\perp$, contingent on whether their orientation is parallel or perpendicular to the local electric (*E*-) field in the metasurface resonant mode excited by the incident waves.

One of the most important design parameters for LC metasurfaces is the thickness of the LC layer; that is, a thicker layer results in a higher drive voltage and slower response time for switching the LC orientation. In scenarios where a moving receiver passes through a sharp beam formed by a large-aperture metasurface in less than one second, the response time must be on the order of milliseconds to track the receiver; for this purpose, an LC layer with a thickness comparable to that of LCDs is preferable. Spherical particles or photosensitive column spacers are commonly used as spacers to determine the LC thickness. Given the inability to control the dispersion of spherical particles, it is preferable to use a photosensitive column spacer whose position can be pre-designed to obtain the desired LC metasurface characteristics. The existing LCD fabrication processes typically make displays with column spacer heights of approximately 3.5 μm[31–33]. Thus, for

[1]Device Technology Labs., NTT, Inc., Atsugi, Japan. [2]Department of Electrical and Computer Engineering, Yokohama National University, Yokohama, Japan. ✉ e-mail: daisuke.kitayama@ntt.com

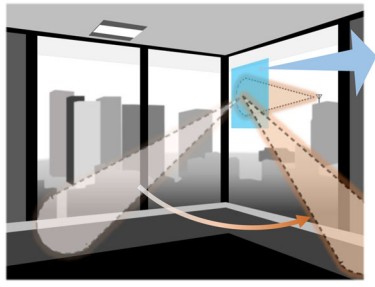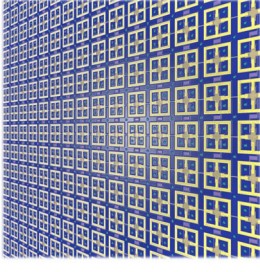

**Fig. 1 | Conceptual illustration of a transmissive liquid-crystal metasurface used as a reconfigurable intelligent surface in mobile communication systems.** The subteraherz band, which is being considered for use in next-generation mobile communication systems, exhibits weak diffraction effects, thereby presenting a challenge for non-line-of-sight (NLoS) communications in an outdoor-to-indoor environment. By situating a transmissive metasurface with an area larger than the 1st Fresnel radius at the boundary between outdoors and indoors (e.g., a window of a building), a controllable propagation path can be formed, and coverage to NLoS areas becomes possible.

compatibility with these processes, the LC metasurfaces should have comparable thicknesses.

There are several reports on reflective and transmissive LC metasurfaces[21–30]. In the reflective type, there are a few reports that achieve a small distance of less than 10 μm between the top and bottom metal in the LC-filled space. Here, the metasurface had a non-resonant wave guiding structure with a large unit cell and multiple metal layers to control the phase of reflected waves[29,30]. On the other hand, the studies on the transmissive type that can be installed on a window to form a propagation path between indoors and outdoors (Fig. 1) do not indicate that the LC layer can be made thin enough and can accommodate the linearly polarized waves used in mobile communications[34–44]. Furthermore, the formation of orthogonal biaxial bias lines with low loss for two-dimensional dynamic beamforming remains a major challenge for transmissive metasurfaces because of the unavoidable interaction between the incoming waves and the bias lines.

Here, we report a transmissive LC metasurface that enables two-dimensional dynamic beamforming at subterahertz frequencies using a single LC layer with a thickness of 3.5 μm. The proposed unit-cell structure is based on a stepped split-ring resonator (SRR) that concentrates the local $E$-field of the resonant mode excited by the incident waves within the LC layer without strong magnetic-dipole confinement, allowing the resonance characteristics to be controlled independently of unit-cell scaling. This design flexibility allows metasurfaces with display-compatible LC thickness to be extended across the microwave to subterahertz frequency range. A large-area metasurface consisting of 47,524 unit cells was fabricated and experimentally characterized at 115 GHz. Using amplitude modulation and a line-matrix control scheme, the metasurface demonstrates two-dimensional beam steering and variable focusing with a 3-dB bandwidth of approximately 10%, while supporting dual linear polarizations. These results indicate that LC metasurfaces with display-compatible thicknesses can provide practical wavefront control for next-generation high-frequency wireless communication systems.

## Results

The most straightforward configuration of an LC metasurface is a bilayer unit resonator with an LC layer between them. Applying a voltage between the stacked resonators generates an $E$-field within the LC of the region where the resonators overlap. However, this configuration is problematic when the LC layer is thin. Here, we conducted electromagnetic simulations (HFSS, ANSYS Inc.) on a unit cell structure in which an LC with $\varepsilon_{r\parallel}$ of 2.5 and $\varepsilon_{r\perp}$ of 3.5 was sandwiched between a bilayer cross-dipole patterned with copper. In this simulation, a rotation of the LC orientation was reproduced by modifying the dielectric constant of the LC in the overlap region. The bilayer cross dipole has two hybridized modes: a symmetric mode in which electric

dipoles (ED) in the same direction between layers are excited, and an antisymmetric mode in which EDs in the opposite direction are excited to form magnetic dipoles (MD) at higher and lower frequencies, respectively (Fig. 2a). The symmetric mode causes currents in the two layers to be in the same direction, which results in repulsive forces within the system, and it is a high-energy (high-frequency) mode. Conversely, the antisymmetric mode causes currents in the two layers to be in opposite directions, which results in attractive forces within the system, and it is a low-energy (low-frequency) mode[45,46]. In the symmetric mode, the local $E$-field spreads to regions outside the LC layer, resulting in poor resonant-mode control due to changes in the LC orientation. In the antisymmetric mode, the local $E$-field is concentrated in the LC layer because of the excitation of charges with opposite signs in each layer, and the resonance frequency shifts markedly with the LC orientation change (Fig. 2b). As the thickness of the LC layer decreases, the symmetric mode becomes more repulsive between the layers of the ED and shifts to a higher energy while maintaining the resonance peak (Fig. 2c). On the other hand, in the antisymmetric mode, the resonance frequency shifts to the lower frequency side, and the peak intensity decreases with decreasing LC thickness. In the simulation, the peak became undetectable when the LC thickness ($t_{LC}$) was 30 μm or less. This is because the strong confinement of the MD leads to an increase in the quality factor (Q) of the antisymmetric mode, resulting in a collapse of the high-Q peak due to ohmic or dielectric losses (Supplementary note 2). These findings indicate that controlling the resonance modes of a simple bilayer resonator with an LC that is much thinner than the wavelength is difficult, irrespective of the mode employed.

In light of the above simulations, we decided to design a stepped SRR (S-SRR) in which the $E$ field is concentrated in the LC layer without MD confinement (Fig. 2d). The S-SRR consists of ring-shaped top metal and cross-shaped bottom metal, with a gap formed perpendicular to the meta-surface plane. This structure has a circulating-current mode similar to that of conventional SRRs, and the $E$ field is concentrated in a 3.5-μm-thick LC (Fig. 2e). In the circulating-current mode of the S-SRR, the absence of a loop current path in the cross-section results in there being no MD excitation between the top and bottom metal layers (Supplementary Note 3). The ring part in the S-SRR corresponds to an inductance component, $L$, and the gap part corresponds to a capacitance component, $C$, which forms the $LC$-resonant circuit (Fig. 2f). When the LC is oriented parallel to the metasurface plane, the dielectric constant of the LC, which determines the capacitance component of the S-SRR, is $\varepsilon_{r\perp}$. When a voltage is applied to the gap and the LC orientation becomes parallel to the local $E$-field in the resonant mode excited by the incident waves in the gap, the dielectric constant changes to $\varepsilon_{r\parallel}$. This structure allows the LC thickness to be determined independently of the cell scaling by properly designing the overlap area at the gap. Therefore, LC metasurfaces with a 3.5-μm-thick LC can be used for not only the over-100-GHz band but also the 28-GHz bands of fifth-generation mobile communication systems (5 G) or 10-GHz bands discussed for sixth-generation mobile communication systems (6 G) (Fig. 2g, i). For example, the S-SRRs we designed for the 10-GHz band have a thickness less than $\lambda/8333$ and have a transmittance modulation greater than 25 dB (Fig. 2g). Furthermore, the four-fold rotational-symmetry structure of the S-SRRs can accommodate both vertical and horizontal polarizations.

To achieve two-dimensional beamforming, it is necessary to route the control bias lines in two orthogonal directions. The presence of conductors longer than the wavelength with components parallel to the $E$-field of the incoming waves affects the interaction of the designed elements with the waves. Here, we simulated the transmission characteristics of $y$-polarized waves incident on the following structures: $x$- and $y$-directional bias lines with a sheet resistance of 0.86 Ω per square (0.86 Ω/sq) (Fig. 3a), S-SRRs (Fig. 3b), and S-SRRs connected to bias lines in the $x$ and $y$ directions (Fig. 3c). The material parameters except for the bias lines were set to be ideal (PEC for the S-SRR pattern and a loss tangent of 0 for the glass and LC layer), while the sheet resistance of the 25-nm-thick bias lines with a linewidth of 5 μm was varied from 0.86 to 800 Ω/sq. As shown in Fig. 3d, the resonance peak of the transmittance decreases as the sheet resistance of the bias line decreases. When the sheet resistance is that of copper, 0.86 Ω/sq, the

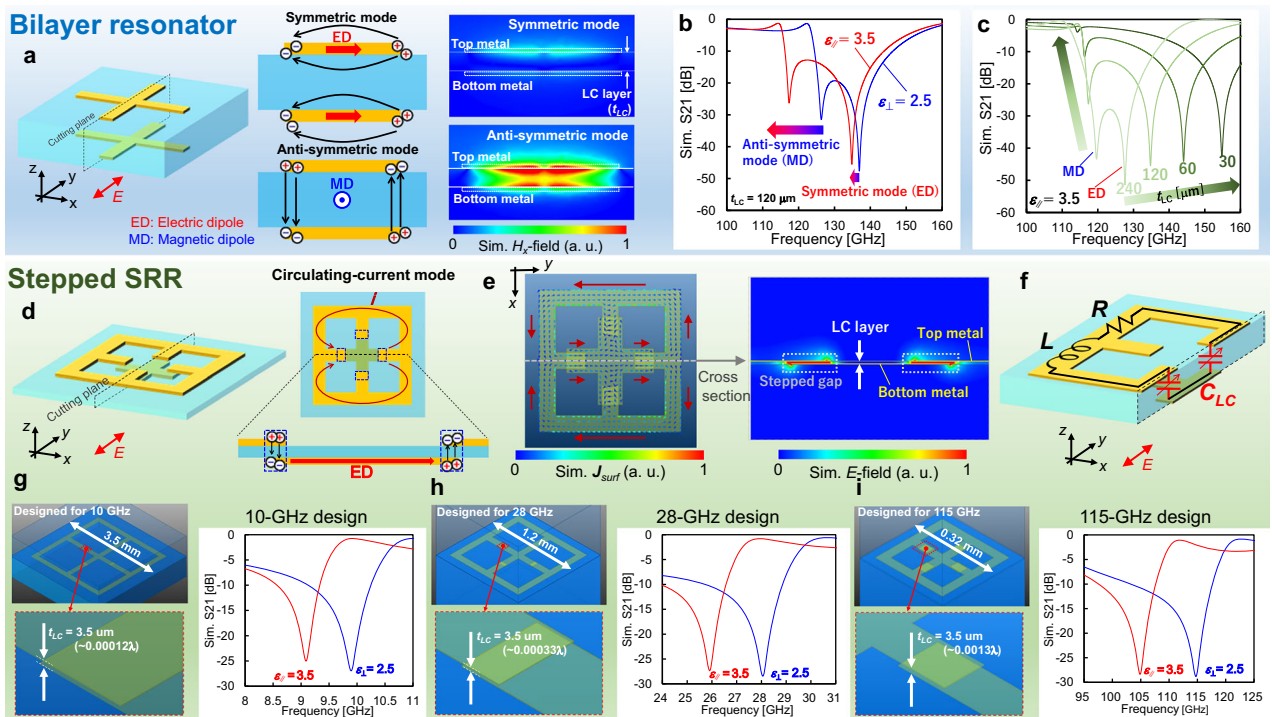

**Fig. 2 | Bilayer resonators and proposed stepped split-ring resonators (S-SRRs) for liquid-crystal (LC) metasurfaces with a thin LC layer. a** Schematic diagram of the bilayer resonator and its symmetric and antisymmetric modes. Simulated distribution of surface current density. The Hx component of the H-field indicates the formation of the magnetic dipole (MD) between the layered resonators in the antisymmetric mode. **b** Simulated transmittance assuming a LC layer with a dielectric constant of a 2.5 for εr⊥ and 3.5 for εr∥. In the symmetric mode, wherein the local E-field spreads outside the LC layer, the resonance peak exhibits a minimal shift even when the dielectric constant of the LC undergoes a change. **c** Simulated transmittance of the bilayer resonator exhibiting a decrease in resonance peak intensity as the thickness of the LC layer decreases in the antisymmetric mode. The

resonance peak is nearly undetectable when the thickness of the LC is less than 30 μm. **d** Schematic diagram of the S-SRR and its circulating-current mode. In this resonance mode, charges with opposite signs are excited in the gap between the top and bottom metals. **e** Simulated distribution of the surface current density and the E-field demonstrating that the circulating current is indeed excited, and that the local E field is concentrated at the gap. **f** Schematic diagram of the S-SRR and its equivalent circuit for the circulating-current mode. **g–i** Schematic diagrams and simulated transmittance of S-SRRs designed for the 10-, 28-, and 115-GHz bands. All designs with a constant LC thickness of 3.5 μm could control the resonance peak. (see detailed dimensions in Supplementary Note 13).

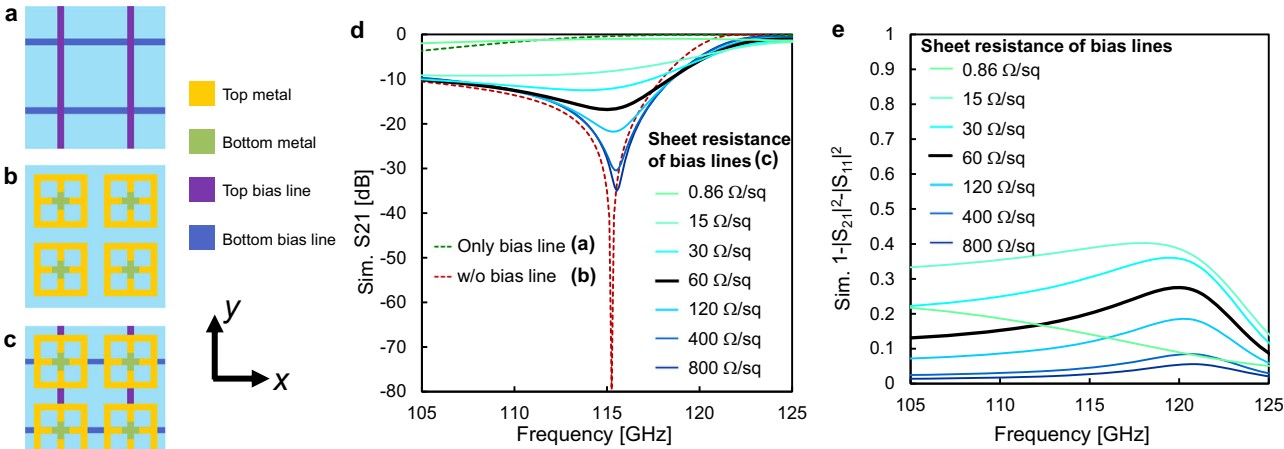

**Fig. 3 | Effect of biaxial bias lines and their material on transmission properties.** Schematic diagrams of the simulated unit cells of **a** x- and y-directional bias lines, **b** stepped split-ring resonators (S-SRRs), and **c** S-SRRs connected to the bias lines. **d** Simulated transmittance and **e** power dissipated in the bias lines connected to S-

SRRs, in which the sheet resistance of the bias lines is varied, and the other material parameters are set to be ideal. Bias lines with low sheet resistance strongly affect the scattering characteristics of the S-SRRs. In contrast, bias lines with relatively high impedance preserve the original characteristics of the S-SRRs.

characteristics of the bias lines become dominant, and the resonance peak of the S-SRRs is no longer evident. The S-SRR's small unit cell size of less than $\lambda/8$ and its line width of only a few tens of micrometers in the subterahertz frequency band present a challenge in loading a stub to choke an RF signal

with a much narrower line width than the S-SRR over a large area. Thus, we designed a unit cell with two distinct types of conductive material: copper for the part intended to interact with incoming waves and indium tin oxide (ITO) for the bias lines, wherein ITO has higher resistance (60 Ω/sq in this

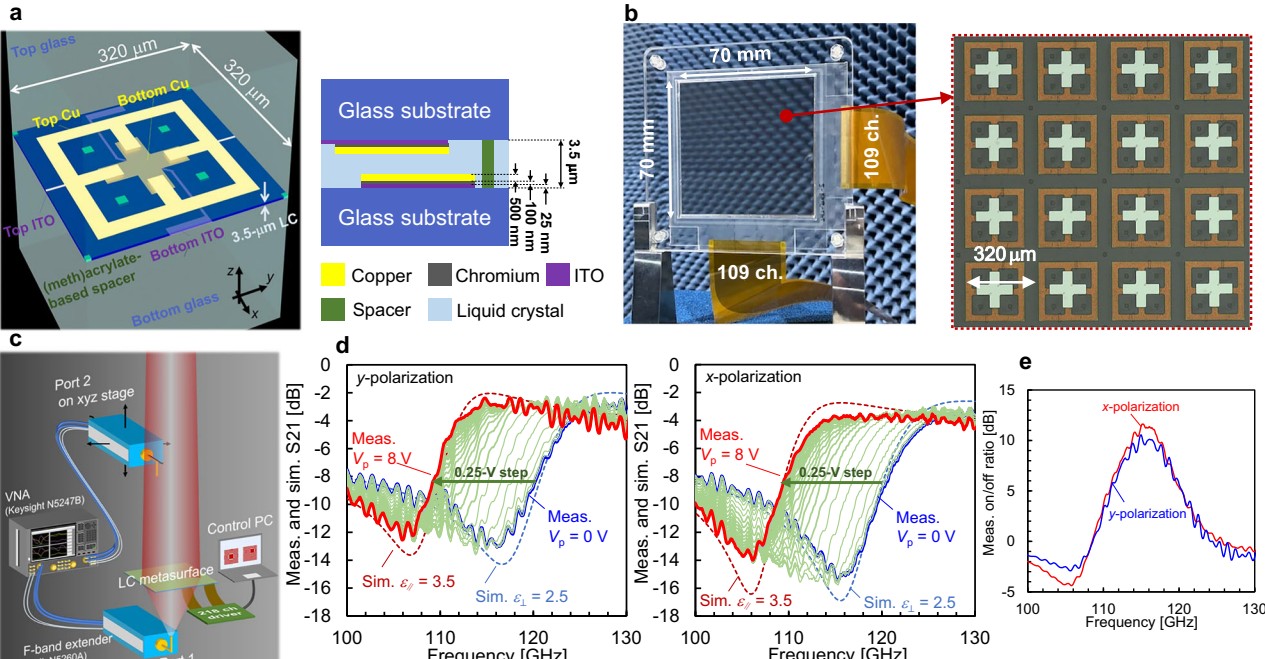

**Fig. 4 | Fabricated liquid-crystal (LC) metasurface and its basic characteristics.** **a** Schematic images of stepped split-ring resonators (S-SRRs) cell designed for the 115-GHz band and its layer structure (see detailed dimensions in Supplementary note 13). **b** Photographs of the fabricated LC metasurface and magnified view. The flexible printed-circuit cables (109 ch) were connected to the bias lines on the top and bottom glass substrates. The indium tin oxide (ITO) bias lines are transparent to visible light and thus are not visible in the magnified photograph. **c** Experimental configuration of near-field distribution measurements using vector network analyzer (VNA) (see the "Methods" section). **d** Measured transmittances for incident waves of $y$- and $x$-polarization, in which the peak voltage of the square waves is varied from 0 to 8 V. **e** The modulation depth of the transmittance between the off-state of 0 V and on-state of 8 V.

study) than copper and high compatibility with the LCD manufacturing process. As shown in Fig. 3d, this configuration enables a resonance peak of greater than 10 dB to be maintained even when two-axis bias lines are connected to the S-SRRs. Figure 3e shows the power $(1- |S_{21}|^2 - |S_{11}|^2)$ dissipated in the bias lines connected to the S-SRR. The dissipated power increases as the sheet resistance increases, and the peak dissipated power is slightly higher in frequency than the peak in transmittance. The dissipated power at 115 GHz is 20%, which is large compared to that of the glass substrate, LC layer, or metal pattern (Supplementary note 4), indicating that reducing the power dissipated in the bias line is the most effective way to enhance the modulation depth of the S-SRR.

Figure 4a shows the S-SRR designed for the experimental verification in the 115-GHz band and its layer structure. The fabricated LC metasurface comprised 47524 (218 × 218) cells in a 70 mm square area with a cell size of less than 320 μm square (<λ/8) and used two bias lines together as a control unit (109 channels in each of the $x$ and $y$ directions; Fig. 4b). Since the aperture of the metasurface, which was assumed to be placed in the propagation channel, should be a few orders of magnitude larger than the wavelength, direct far-field measurements were difficult; therefore, near-field distributions were evaluated instead. The measured results were then compared with far-field patterns and corresponding near-field distributions calculated from a channel model of the metasurface (equation S14 in Supplementary Note 5). For the far-field validation of the fabricated LC metasurface, comparison of measured and calculated far-field distributions using a reduced aperture is provided in Supplementary Note 6.

First, a uniform control signal was applied to all of the S-SRRs to ensure that they operated as designed. Here, it should be noted that all the experiments were performed at room temperature, 25 °C. When the peak voltage ($V_p$) of the square wave was 0 V, the transmission characteristics were consistent with the simulation result for the LC dielectric constant of $\varepsilon_{r\perp} = 2.5$, and the resonance peak shifted to the lower frequency side as $V_p$ increased (Fig. 4d). The shift in the resonant peak was small when $V_p$ was 4 V or higher, and at 8 V, no change could be observed. The peak at $V_p$ of 8 V

matched the simulation result for the LC dielectric constant of $\varepsilon_{r\parallel} = 3.5$, demonstrating that the S-SRRs worked as designed. The modulation depth of the transmittance for $V_p$ values from 0 to 8 V was approximately 12 dB (Fig. 4e). The slight discrepancy in the transmission characteristics of the $x$- and $y$-polarized incoming waves can be attributed to the disparate lengths of the control bias lines between the top and bottom layers (Supplementary note 7 provides the interpolation between orthogonal polarizations and oblique-incidence data). Here, the states in which $V_p = 0$ and 8 V are applied to the S-SRR are referred to as off and on states, respectively, where the modulation depth saturates and becomes insensitive to $V_p$ changes. The entire LC metasurface between these states could be used, for example, as a switch for areas in which communication is possible.

Next, we performed dynamic beamforming by using amplitude modulation to control the two-dimensional intensity profile of the transmitted waves. The main reason for employing amplitude modulation is to realize a transmissive-type RIS with a single LC layer. The phase change in the waves scattered by a single resonant mode is theoretically limited to π. Here, in contrast to reflective metasurfaces, the waves from a transmissive metasurface consist of both scattered and non-scattered waves, and the non-scattered waves do not undergo phase changes due to resonance modes. This means that a multilayer structure would be needed in order to achieve controllability over a 2π range. Amplitude modulation can form a 1-bit (1/0) discrete wavefront with a single LC layer, which results in low design complexity and fabrication cost. The control method was a line matrix control, which selects signals on a column-by-column and row-by-row basis, rather than an individual matrix control, which selects signals on a cell-by-cell basis; thus, the wavefront of the transmitted wave only approximates an ideal wavefront. This line matrix control has the advantage of reducing the number of control channels: it requires only 109 + 109 = 218 channels for the fabricated metasurface (Supplementary note 8), whereas the individual matrix control requires 109 × 109 = 11881 channels. If we assume a total propagation distance of 100 m, the size of the RIS should be approximately 200 mm square, and the cells would number in

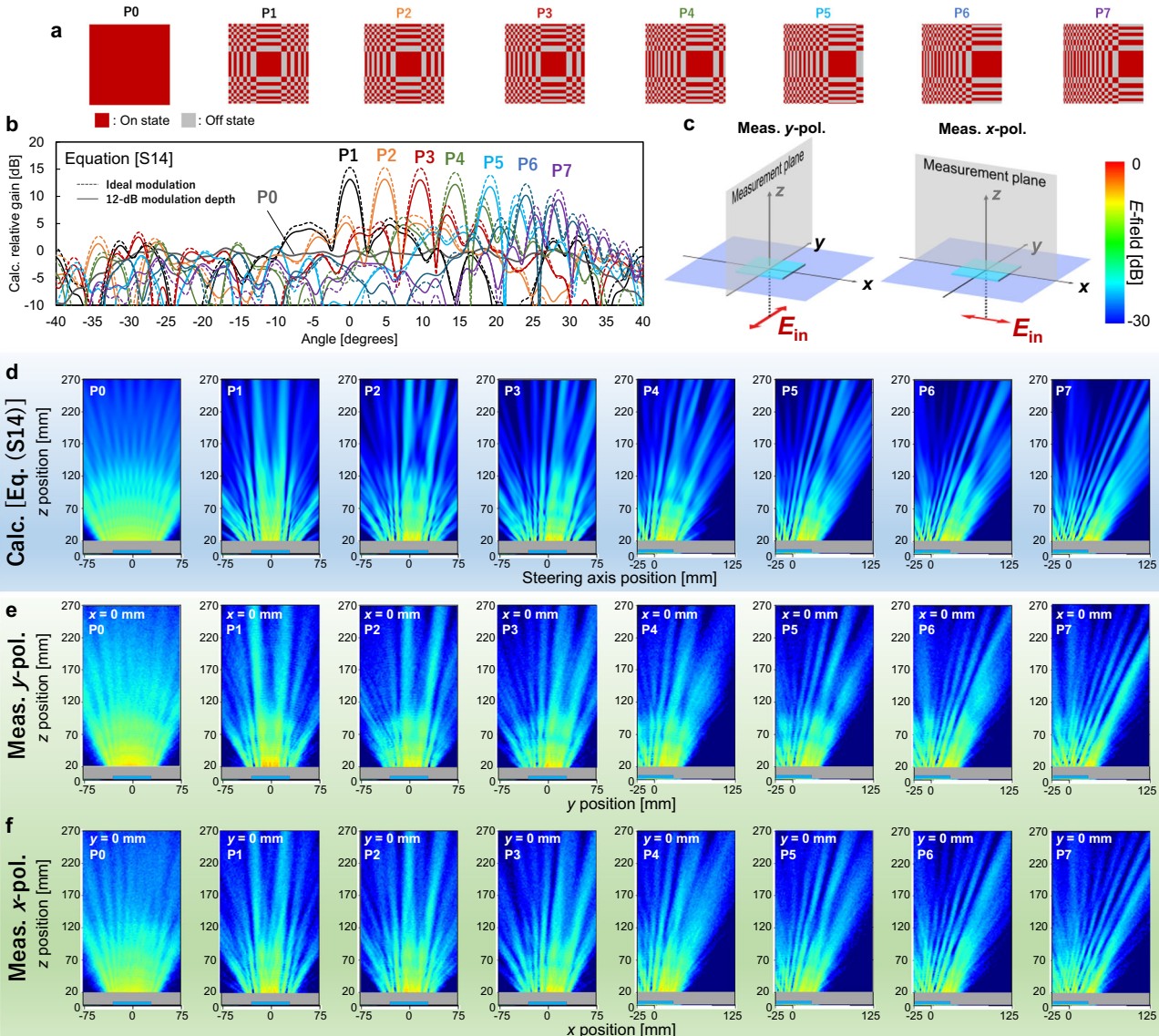

**Fig. 5 | Measured and calculated beam-steering characteristics. a** Control patterns: P0 with all cells in the on-state, P1 to collimate the arriving waves in two dimensions, and P2 to P7 to steer the collimated waves from 0° to 30° in the measurement plane. **b** Calculated far-field patterns for control patterns P0 to P7, which were normalized by the value of P0 at 0°. **c** Configuration of measurement plane, which is the steering plane when measuring with $y$-polarized and $x$-polarized incident waves.

**d** Calculated $E$-field distributions for control patterns P0 to P7. The results of these calculations do not account for polarization and are employed as a reference for both the $x$- and $y$-polarization measurements. **e** Measured $E$-field distributions for $y$-polarized waves in the $yz$ plane. **f** Measured $E$-field distributions for $x$-polarized waves in the $xz$ plane.

the hundreds of thousands. Thus, the line-matrix control is a reasonable method from the perspectives of control complexity and driver cost for adaptively controlling LC metasurfaces. Furthermore, it does not need thin-film transistors (TFTs) to be integrated, as is done with LCDs, so it also has an advantage in terms of lower manufacturing costs. For example, a reflection-type programmable metasurface, in which the two-dimensional line-by-line control pattern is generated through the modulo-addition of column and row coding matrices, was reported to demonstrate two-dimensional beam steering without integration of the TFT layer[47]. It should be noted that finding the optimized modulation profile is an NP-hard discrete optimization problem[48], and the profile employed in this study is not guaranteed to be the optimal discrete control.

The distributions of the on/off states used in the measurements are illustrated in Fig. 5a. Control pattern P0, with all cells in the on-state, transmits the incident wave as is, whereas P1 collimates the incident wave in both the x and y directions. Patterns P2–P7 steer the collimated wave from 5° to 30° in 5° increments. In most cases, it is necessary to change the

direction and focus of the transmitted wave by using amplitude modulation in which on-state and off-state cells coexist, except for the case where the Tx, RIS, and RX are in a straight line, and their relative position is such that the RIS is smaller than the Fresnel zone (Supplementary note 9). The calculated far-field beam patterns for P0 to P7, which were normalized by the value of P0 at 0°, are shown in Fig. 5b. Here, each cell was assumed to have a modulation depth of 12 dB or infinity. The gain for P1 was 13 dB greater than that for P0 because of the two-dimensional collimation for the modulation depth of 12 dB, and it was 2-dB lower than the gain for the ideal (infinite) modulation depth. Moreover, the gain of the collimated waves decreased as the steering angle increased. This result can be attributed to the decrease in the effective aperture or the increase in the spatial frequency of amplitude modulation for larger steering angles. When the ideal wavefront is discretized or approximated in the metasurface plane, reducing the size of the unit cell decreases the discrepancy between the formed and ideal profile and improves steering accuracy (Supplementary note 9). The S-SRR has a small cell size, ~$\lambda/8$, through its use of circulating-current mode and has

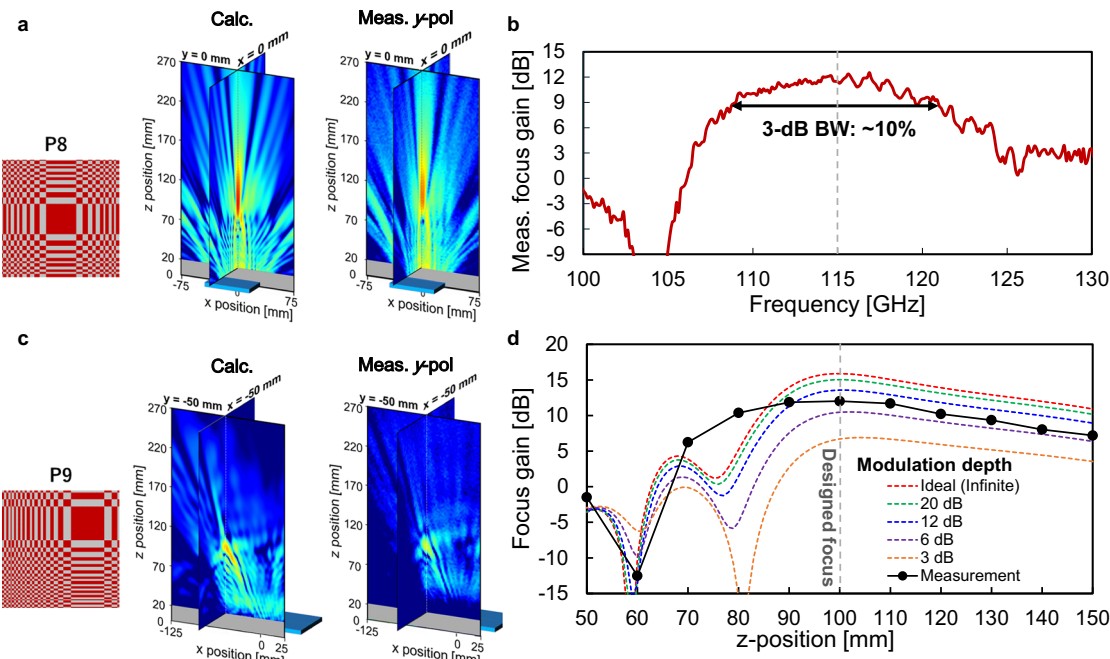

**Fig. 6 | Measured and calculated dynamic focus characteristics. a** Control pattern P8 and calculated and measured *E*-field distributions in the *E*- and *H*-planes. **b** Measured frequency characteristic of lens gain, defined as the gain relative to the result for P0, at the focus (*x*, *y*, *z*) = (0, 0, 50) mm for control pattern P8. **c** Control pattern P9 and calculated and measured *E*-field distributions in the *E*- and *H*-planes. **d** Measured and calculated focus gain on the central axis for P8, in which the modulation depth in the calculations is varied from 3 dB to an ideally infinite value.

more potential compared with other structures to achieve good steering gain and accuracy at large angles. The calculated near-field distributions on the steering plane (Fig. 5c) corresponding to each far-field pattern from P0 to P7 are presented in Fig. 5d. The near-field distribution also corroborates the observation that the spreading wave is collimated and steered through angles corresponding to the far-field beam direction. Note that equation (S14) does not take polarization into account; thus, the calculated results in the steering plane will be identical regardless of whether the steering direction is in the *x*- or *y*-direction. With the calculated results as a reference, we measured the steering behavior in the *yz* plane with the *y*-polarized incident wave (Fig. 5d). In the case of P1, the transmitted wave was collimated in the *y* direction as well as in the calculated results and was similarly collimated in the *x* direction (Supplementary note 11), thereby achieving a two-dimensional wavefront control. The measured near-field distributions for the other control patterns were also consistent with the calculated results, demonstrating that the LC metasurface with S-SRRs can dynamically control transmitted waves in two dimensions. The measurements conducted on the *xz* plane with *x*-polarized incident waves are also in accordance with the calculations, indicating that the S-SRR is capable of accommodating both vertical and horizontal polarizations (Fig. 5e).

Furthermore, S-SRRs can be used to regulate the near-field distribution because their dimensions are less than *λ*/4, even when they are controlled in 2×2 cell units. The transmitted waves were focused at (*x*, *y*, *z*) coordinates of (0, 0, 100 mm). Control pattern P8 resulted in the transmitted wave being concentrated at the designed focus in both the *xz* and *yz* planes (Fig. 6a). A lens gain of 12 dB relative to P0 and a 3-dB bandwidth of approximately 10% were achieved at the focus (Fig. 6b), which is sufficient for mobile communication systems. The two-dimensional focusing operation was also validated with control pattern P9 (Fig. 6c) when the focus was moved off the central axis of the LC metasurface by 50 mm in the *x* and *y* directions (−50, −5, and 100 mm). Figure 6d shows the measured and calculated focus gain on the central axis for control pattern P8, in which the modulation depth in the calculation was varied from 3 dB to ideally infinite. The larger the modulation depth was, the higher the focus gain became at the designed focus of *z* = 100 mm. Similarly to Fig. 5b, the calculated focus gain at 12-dB modulation depth was approximately 2 dB lower than that of the ideal

infinite modulation depth. If the modulation depth can be improved to 20 dB, this gain degradation can be suppressed to less than 1 dB, which indicates that further optimization of bias lines, including their sheet resistance and design, would be effective in improving the lens gain, as shown in Fig. 3d. Although there is a discrepancy of 1.5 dB between the measured and calculated focus gain in Fig. 6d, the expected results can be obtained by considering the effect of the finite aperture of the probe antenna used in the experiment and the measurement error due to unwanted reflections in the experimental system.

## Discussion

We demonstrated two-dimensional dynamic beamforming of transmitted waves using a metasurface with a 3.5-μm-thick LC. In this study, amplitude modulation of 1 bit (1/0) was used to control a two-dimensional intensity profile of the transmitted waves in the metasurface plane. If the applied voltage between the top and bottom metal is multibit, or greyscale, instead of binary, it is possible to make a greyscale intensity profile. In the aperture with phase modulation, the deviation of the quantized phase distribution from the ideal continuous phase distribution can be reduced by implementing greyscale control, which would increase the diffraction efficiency of first-order diffracted waves $\eta_1$, expressed as

$$\eta_1 = \mathrm{sinc}^2\left(\frac{1}{N}\right) \qquad (1)$$

where $\mathrm{sinc}(x) = \sin(\pi x)/(\pi x)$, and $N$ is the quantization level in $2\pi$ phase[49]. On the other hand, for the amplitude-modulated aperture, the cell state is determined to be transparent or opaque on the basis of whether the waves transmitted through each cell constructively interfere at the desired point. Therefore, considering only maximization of the received power, there is no benefit from greyscale control. On the other hand, given the correlation between the intensity profile in the aperture plane and the transmissive beam pattern including sidelobes, implementation of a greyscale control could have the effect of suppressing sidelobes. In scenarios where the interference with other wireless systems has to be suppressed, it is worth considering greyscale control even for amplitude modulation.

## Table 1 | Comparison of the proposed cell and other transmissive liquid-crystal (LC) metasurfaces

| Ref. | Frequency [GHz] | Structure | Polarization | LC thickness [μm] | Wavefront modulation | Scanning capability | Cell size (Array size) | Measured insertion loss [dB] |
|---|---|---|---|---|---|---|---|---|
| 34 (Meas.) | 27.5 | Fishnet | Linearly | 762 × 2 layer (0.71λx2) | Phase (~180°) | 1-D | 0.42λ (8 × 8) | N.A. |
| 35 (Meas.) | 820 | Split-ring resonator (SRR) | Linearly (Conversion) | 800 (2.2λ) | Amplitude (~30 dB) | Intensity control | 0.16λ (N.A.) | ~14 |
| 36 (Meas.) | 9.75 | Dipole | Linearly | 600 (0.02λ) | Phase (~90°) | 1-D | 0.36λ (15 × 9) | ~25 |
| 37 (Sim.) | 340 | Patch | Linearly | 580×2 layer (0.65λx2) | Phase (~180°) | 1-D | 0.38λ (20 × 20) | N.A. |
| 38 (Meas.) | 793 | Wires | Linearly | 500 (1.32λ) | Phase (~90°) | N.A. | 10.5λ (N.A.) | N.A. |
| 39 (Meas.) | 690 | SRR | Linearly (Conversion) | 250 (0.58λ) | Amplitude (N.A.) | Intensity control | 0.23λ (150 × 150) | N.A. |
| 40 (Meas.) | 528 | SRR | Linearly | 50 (0.088λ) | Amplitude (~6 dB) | Intensity control | 0.1λ (N.A.) | ~19 |
| 41 (Meas.) | 345 | Patch | Linearly | 50 (0.058λ) | Amplitude (~15 dB) | 1-D | 0.89λ (40 × 40) | ~1.2 |
| 42 (Meas.) | 421.2 | Ring slot | Dual-linearly | 45 (0.063λ) | Amplitude (~30 dB) | Intensity control | 0.43λ (46 × 46) | 1.19 |
| 43 (Meas.) | 800 | Meandering wires | Dual-linearly | 12 (0.032λ) | Phase (~14°) | 2-D | 0.22λ (10 × 10) | ~9.5 |
| 44 (Meas.) | 426 | SRR based | Linearly | 60 (0.085λ) | Phase (~180°) | 1-D for prototype | 0.45λ (48 × 48) | N.A. |
| Ours (Meas.) | 115 | SRR based | Dual-linearly | 3.5 (0.0013λ) | Amplitude (12 dB) | 2-D | 0.12λ (218 × 218) | ~2.5 |

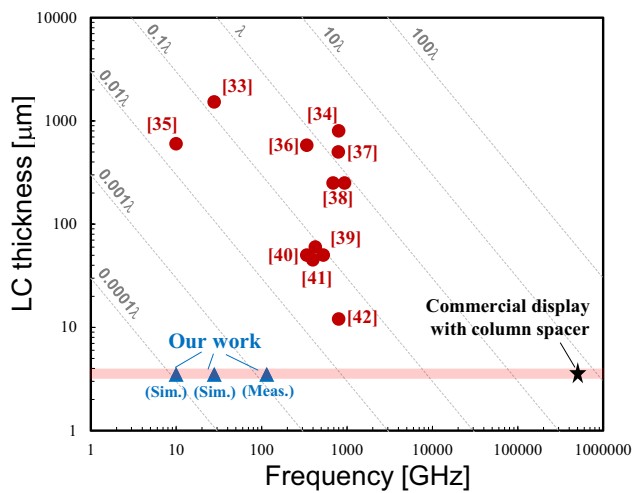

**Fig. 7 | Liquid-crystal (LC)-thickness benchmark of the proposed transmissive LC metasurface.** The proposed design of the stepped split-ring resonator has a remarkably thin LC thickness of 3.5 μm.

Table 1 compares other published transmissive LC metasurfaces and the proposed design in terms of polarization, LC thickness, wavefront modulation, scanning capability, cell size, and insertion loss. In this table, insertion loss refers to the experimentally measured transmission loss for a uniformly controlled cell array. For amplitude-modulation metasurfaces, the loss value corresponds to the transmission loss at the operating frequency when the cell is in its lowest-loss state (on-state in our case). For phase-modulation metasurfaces, it represents the lowest-loss state among the available phase configurations. The expected response time of the proposed S-SRR was evaluated using a sample with the same LC thickness (3.5 μm) as the proposed structure, exhibiting a rise time of 3 ms and a fall time of 50 ms at room temperature, 25 °C (Supplementary note 12). There have been a few reports that have evaluated the response times of transmissive metasurfaces. Therefore, this table compares the LC thicknesses, which are strongly correlated with the response times of the LC orientation changes. Given that the response time slows down in proportion to the square of the LC thickness[50], an LCD level of thickness would be important not only for compatibility with the LCD fabrication process, but also for achieving a fast response. Figure 7 summarizes the LC thicknesses of the proposed S-SRR, and other reported transmissive LC metasurfaces, highlighting the remarkably thin LC layer of 3.5 μm in our design. Regarding diffraction efficiency, there have been no reports evaluating the efficiency of transmissive LC metasurfaces; therefore, the wavefront modulation scheme is included in the comparison. According to equation (1), the efficiency with phase modulation depends on the quantization number of the phase range of 2π; it is 41% for binary modulation and 81% for 2-bit modulation. For transmissive metasurfaces, a multilayer structure is needed in order to achieve good phase controllability, and the reported transmissive LC metasurfaces with phase modulation employ multiple LC layers. Although the theoretical diffraction efficiency of amplitude modulation is 10%[51], it can be achieved with a single LC layer, a feature that is advantageous in terms of design complexity and fabrication cost. Most of the reports have demonstrated uniform control or one-dimensional beam control. When these controls are extended to two-dimensional control, losses due to bias lines should be considered. In the proposed cell, control bias lines in orthogonal directions are connected to each cell to achieve two-dimensional beamforming, and they were experimentally demonstrated to have a low insertion loss of approximately 2.5 dB for the largest (218 × 218) array scale. Furthermore, our design has a four-fold rotational symmetry, including the bias lines, which supports both two-dimensional beamforming and dual linear polarization.

An effective approach to maximize the performance of the S-SRRs and achieve the theoretically highest efficiency is to enhance the modulation depth by reducing the dissipated power in the bias lines, as was shown in the Results section. A promising alternative way of achieving a large modulation depth, particularly at higher frequencies such as the terahertz band, where the ohmic loss of metals becomes increasingly important, is the use of dielectric-based resonators. Recently, dielectric approaches to achieving quasi-bound states in continuum resonances with a high quality factor in the near-infrared spectrum have been reported[52,53]. In the reported BIS resonances, the processing accuracy of the cell structure remains problematic. However, in the terahertz band, it should be possible to fabricate dielectric-based resonators with sufficiently fine accuracy relative to the wavelength. If the required dielectric thickness in proportion to the operation wavelength is not a problem, the combination of using BIC resonances and LC materials would be a promising approach to making a large-scale LC RIS with a large modulation depth.

In this study, the S-SRR was used to make an amplitude-modulated transmissive metasurface. It should be noted, however, that the S-SRR can also be applied to reflective phase-modulation metasurfaces. Most reported reflective metasurfaces also utilize resonance shifts, and the use of the S-SRR, which can achieve resonance shifts with an LC thickness of 3.5 μm over a wide frequency range, as a base structure would enable industrialization of large-area reflective RISs.

We also used a line-matrix control in order to reduce the number of control channels and fabrication cost. Here, if an increase in the number of control channels and the cost is acceptable, integration of TFTs into LC RISs is a promising approach, allowing for the formation of more ideal wavefronts through pixel-level matrix control. The additional TFT layer would cause losses, particularly in the transmissive type; it would be crucial to identify the optimal control method by comparing the efficiency gain of the improved wavefront with the insertion loss of the TFT layer.

To control the characteristics of the propagation channel through the use of metasurface devices, it is essential that these devices be of a size comparable to the 1st Fresnel zone at their installation position. Moreover, in light of the impact of the devices on the landscape, particularly in scenarios such as their installation on a window, as illustrated in Fig. 1, it is desirable for them to be optically transparent. To enhance transparency, the following two approaches could be considered. One is to use low-resistance ITO for the S-SRR pattern and high-resistance ITO for the control bias lines. Given the higher transparency of low-resistance ITO in comparison to the copper used in this paper, an enhancement in RIS transparency can be expected as well. However, it should be noted that this approach increases the sheet resistance of the S-SRR pattern, resulting in a trade-off between transparency and modulation depth. The other way is to mesh the metal pattern[5], excluding the overlapping area, which is a crucial part for maintaining the controllability of the S-SRR. Since the overlapping area constitutes a small percentage of the current metal pattern area, meshing areas other than the overlapping part is expected to enhance the optical transparency.

To ensure the practical implementation of LC-based RIS, long-term stability under environmental stress must be considered. When installed on a window as an indoor/outdoor interface (Fig. 1), the LC layer, which exhibits absorption in the ultraviolet (UV) band[54], should be protected from UV exposure. This can be achieved by incorporating UV-cut glass, UV-blocking films, or protective coatings, as commonly employed in outdoor LCD applications such as digital signage. In addition, thermal management is important because heat can degrade LC materials[55]. Although RIS does not generate self-emitted heat from a backlight and its thermal environment is relatively less severe than that of outdoor LCDs, passive or active cooling strategies should still be considered to maintain reliability over extended periods. While these mitigation strategies are well established in LCD applications, future work should include accelerated UV and thermal stress testing to quantitatively evaluate the long-term stability of the LC-based RIS under realistic outdoor conditions.

In conclusion, we devised an S-SRR structure with a constant LC thickness that can be easily redesigned by scaling over a wide frequency range (from 10 GHz to over 100 GHz) and used it for dynamic beam-forming of linearly polarized transmissive waves for both vertical and horizontal polarizations. In a simulation, an S-SRR for operation in the 10-GHz band was designed with a display-compatible thin LC layer (less than λ/8333). The fabricated S-SRR for operation in the 115 GHz band had a cell size of less than λ/8 and was shown in an experiment to be capable of beam steering of two-dimensionally collimated transmissive waves by ±30° and focusing with a 3-dB bandwidth of 10%. These findings should facilitate the practical implementation of large RISs and promote the industrial use of subterahertz bands, which are currently being considered for use in 6 G communications and other applications.

## Methods
### Device fabrication
An S-SRR pattern consisting of a 25-nm-thick ITO layer (sheet resistance of 60 Ω square), 100-nm-thick chromium layer, and 500-nm-thick copper layer was formed on each of two 700-μm-thick alkali-free glass substrates (relative permittivity of 5.15). The ITO layer contained a pattern of bias lines and S-SRRs. The chromium layer was formed only where copper was patterned in order to improve the adhesion between the ITO and copper. The glass substrates were laminated using a (meth)acrylate-based column spacer. The patterns on the substrates were faced to each other, and the 3.5-μm-thick space between their substrates was filled with nematic LC material (DHB-012, DIC Corp., detailed information in Supplementary note 14), whose $\varepsilon_{r\perp}$ and $\varepsilon_{r\parallel}$ were 2.65 and 3.84 at 10 GHz. The S-SRR patterns were located on the inside surfaces of the laminated glass substrates; thus, the distance between the surfaces of the top and bottom copper layers was approximately 2.9 μm (Fig. 4a). Note that the alignment accuracy between the top and bottom metal during the fabrication process is an important manufacturing parameter to ensure the reproducibility of the device (Supplementary Note 15).

### Characterization of near-field distribution
The measurement setup was composed of a vector network analyzer (VNA) (N5247B, Keysight Technologies Inc.), VNA extenders for the F-band (N5260A, OML Inc.) at ports 1 and 2, and an xyz linear motor stage with a positioning accuracy of 6 μm (SGSP46, SIGMAKOKI Co., Ltd.) to perform two-dimensional scans. In the experiment shown in Fig. 4, a standard gain horn antenna was used for both ports 1 and 2. In the near-field measurements shown in Fig. 5, waveguide probe antennas were used for both ports, in which port 1 was fixed in position, while port 2 for measuring the transmitted waves was placed on an xyz stage on the other side of the LC metasurface (Fig. 4c). The phase center of port 1 was placed 45 mm away from the LC metasurface on the central axis of the LC metasurface, and port 2 scanned an area of 150 mm in the x or y direction and 200 mm in the z direction in 2-mm steps starting 20 mm away from the LC metasurface. Regarding the control signal generation, a function generator and a 218-channel switch were employed as the driver, facilitating the application of a 1-kHz square wave to any channel of the LC metasurface. RF absorbers were also used to prevent reflections between the instruments. The selection of the channel to which the control signal was to be applied was determined from the distance between the phase centers of port 1 and the position where the transmission waves were to be directed through each cell of the LC metasurface. The assessment of measurement uncertainty was conducted through the implementation of repeated scans, executed under identical conditions within the anechoic chamber. These repetitions confirmed that the uncertainty, including unintended reflections between instruments, remains below ±0.2 dB (Supplementary Note 16).

## Data availability
The detailed dimensions of the fabricated sample's S-SRR cell (Fig. 4b) are in Supplementary Note 13, and the corresponding layout file in dxf format is available from the corresponding author upon reasonable request. The data

that support the findings of this study are available from the corresponding author upon reasonable request.

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

## Acknowledgements
The authors would like to thank S. Nakai for his experimental support. The authors also thank T Sasaki, Y. Nonaka, and S. Hirata at DIC Corp. in Japan for the valuable discussions on LC material.

## Author contributions
D.K. designed the unit cell, conducted HFSS simulations, and conceived the experiment. D.K. and H.K. conducted the experiments and analyzed the results. D.K. and Y.H. conducted the channel calculations. A.P. made contributions to the writing and revision process. H.T. supervised all aspects of the research. All authors reviewed the manuscript.

## Competing interests
D.K., A.P., and H.T. are inventors on a patent application related to this work (WO/2024/095459) filed by NTT, Inc. The remaining authors declare no competing interests.
