## [Transparent Peer Review file · Communications Engineering]

Transmissive metasurface with 3.5- μm -thick liquid crystals for subterahertz-wave dynamic beamforming

Corresponding Author: Mr Daisuke Kitayama

Version 0:

Reviewer comments:

Reviewer #1

(Remarks to the Author)

In this paper, the authors report a split-ring transmissive LC metasurfaces whose LCD-compatible 3.5- μm LC thickness can be determined independently of the scaling of the unit cell and can be redesigned over a broad frequency range from microwave to subterahertz waves. Through 12dB modulation depth of intensity, a prototype shows two-dimensional beam steering and focusing of linearly polarized waves in the 115-GHz band, having a bandwidth of 10% and polarization-independent characteristics. Line-by-line matrix control is used to simplify control.

The work being reported is of good quality, with essential materials displayed to demonstrate its functionalities. To help readers understand, here's a list of specific points the authors should address:

• Technical concerns:

1. In line 54, the authors write "To reduce the total propagation loss of the path through the metasurface to a level commensurate with the free space propagation loss (FSPL) of the total propagation path length, it is necessary that the length of one side of the metasurface be greater than or equal to the radius of the 1st Fresnel zone." Please add references and more reasoning to this comment.
2. In line 67, the authors write "Here, the response time has to be on the order of milliseconds for the LC metasurface to track a moving receiver", please provide a reasoning for this statement as the response time is very scenario dependent.
3. In line 133, the authors write "concentrated in the LC layer without MD confinement (Fig. 2d)", the E-field in Figure 2a asymmetric mode and Figure 2d seem alike, is there also an MD confinement in Figure 2d? What is the difference between these two modes in compressing the LC thickness?
4. In line 180, the authors write "Above the V_p ($= 4\text{ V}$), the peak shift reached a saturation point", and in line 186, "the states in which $V_p = 0$ and 8 V are applied to the S-SRR are referred to as the off and on states", why $V_p = 8\text{ V}$ is needed if the response is saturated at 4 V ?
5. In line 191, "control method was a line-by-line matrix control", is it beneficial to perform greyscale instead of binary control? Is it possible to perform grey-scale line-by-line matrix control?
6. In line 243, the authors write "In the calculations, each cell was assumed to have an ideal modulation that was lossless in the on-state and completely non-transparent in the off-state, while the modulation depth in the measurements was finite at approximately 12 dB", is it possible to incorporate the 12 dB modulation depth in the calculation to confirm the expected experimental performance?
7. In line 384, the authors write "a 100-nm-thick chromium layer, and a 500-nm-thick copper layer", the metals are thick, is it possible to further reduce the metal layer thickness?
8. In line 427, in Figure 2f-h, the overlapping area is redesigned for each frequency band, so the unit cell transparency is also related to the operational frequency band, is there a way to further increase the transparency of the device?
9. In lines 466 and 478, in Figures 5 and 6, what is the maximum and minimum steering efficiency among the scanned angles? Why steering gain or efficiency is low at large angles, and what are the limited factors?

• Description concerns:

1. In line 36, the authors write "usage of the finite frequency resource", as sub-THz also contains finite resources, this description is inaccurate.
2. In line 64, the authors write "is parallel or perpendicular to the local electric (E-) field excited in the metasurface." The excited E-field may refer either to the E-field in the metasurface mode excited by the THz wave or the E-field generated by electric control, which will confuse. This description appears several times in the article, please check and confirm.
3. In line 263, the authors write "works over a wide frequency range (from sub-6 GHz to over 100 GHz)." This statement might be misunderstood as the bandwidth of a single design, need more clarification.

4. In line 427, in Figure 2e, the scale bar of the E-field plot is from 0~1 and is inconvenient to compare with the E-field in Figure 2e, whose scale bar is from -1~1. Please unify the scale bar.

5. In line 439, "f-g" should be "f-h".

6. In line 445, in Figure 3, figure label e is missing.

Reviewer #2

(Remarks to the Author)

The manuscript entitled "Transmissive metasurface with 3.5 μm -thick liquid crystals for sub-terahertz-wave dynamic beamforming" addresses an important bottleneck in 6 G wireless engineering by coupling a liquid-crystal (LC) layer of display-grade thickness with a transmissive reconfigurable intelligent surface (RIS) based on a stepped split-ring resonator (S-SRR). While the general concept is timely and potentially impactful, the theoretical framework and experimental validation leave critical questions unanswered. First, the electromagnetic analysis rests almost entirely on full-wave HFSS simulations; a complementary circuit-level or coupled-mode model is not provided, so the individual roles of capacitance, parasitic inductance and sheet resistance in determining the resonance quality factor and the observed 12 dB modulation depth remain opaque. The paper claims that the antisymmetric mode of a conventional bilayer resonator "collapses" when $t_{\text{LC}} < 30 \mu\text{m}$, yet this assertion is supported only qualitatively (Fig. 2c) without linewidth data, temperature sweeps or loss-tangent measurements that would confirm the Q-factor degradation and its physical origin. Similarly, the mitigation strategy—substituting copper bias lines with ITO—relies on simulated S21 curves but lacks a quantitative assessment of power dissipated in the $60 \Omega/\square$ ITO layer or its thermal stability at 115 GHz. On the experimental side, the near-field maps collected over a 150 mm \times 200 mm scan (Fig. 5e–f) are converted to far-field patterns via equation (S6), yet that conversion assumes perfect periodicity and neglects polarization effects; direct far-field data would be essential to validate the claimed two-dimensional steering accuracy ($\pm 2^\circ$). The reported 12 dB on/off contrast (Fig. 4d–e) is below the ≥ 20 dB generally required to compensate link budgets in indoor RIS deployments, and the authors do not quantify how ohmic losses in copper, dielectric loss in the LC or bias-line impedance non-uniformity each contribute to this limitation. Crucially, the dynamic performance is inferred rather than measured: no rise/fall times or long-term stability tests are reported, yet millisecond-scale switching—and robustness under continuous 1 kHz driving—are central to RIS operation in mobile scenarios. Statistical rigor is also limited: only one 70 mm-square sample, containing 47 524 unit cells, is characterized, and no cell-to-cell variation, wafer-to-wafer reproducibility or measurement uncertainty is discussed. Reproducibility would be greatly enhanced by releasing the GDS layout, full LC permittivity dispersion curves and the driver-electronics schematic. From a broader perspective, the authors position their contribution as "broadband" and "polarization-independent," yet evidence is confined to a 10 % fractional bandwidth around 115 GHz and to two orthogonal linear polarizations; oblique and circular states are not tested, nor are frequencies below 100 GHz where LC loss rises sharply. To contextualize novelty, it would be appropriate to discuss recent all-dielectric approaches that achieve quasi-bound-state-in-the-continuum (BIC) resonances with higher Q and negligible ohmic loss, e.g. J. F. Algorri et al., *Optics & Laser Technology* 161 (2023) 109430, whose silicon-slot metasurface offers an instructive comparison for minimizing insertion loss and enhancing modulation depth. In summary, while the proposed S-SRR/LC platform is promising and could indeed influence future transmissive RIS design, the manuscript requires (i) a more rigorous analytical treatment of resonance physics, (ii) quantitative loss-budget and thermal analyses, (iii) dynamic-response and reliability data, and (iv) expanded bandwidth and polarization measurements before its claims can be considered fully substantiated.

Reviewer #3

(Remarks to the Author)

This paper presents a transmissive metasurface operating in the 115 GHz band that utilizes a thin 3.5 μm liquid crystal (LC) layer for sub-terahertz beamforming. In contrast to conventional approaches that rely on phase control in tunable LC metasurfaces at microwave and millimeter-wave frequencies, the authors propose an amplitude modulation-based control concept.

Strengths:

- The authors have fabricated a functional sample comprising a large number of elements, demonstrating a scalable approach.
- Near-field measurement results are presented.

Weaknesses:

- Several technical aspects of the metasurface implementation and performance are not sufficiently explained.
- A comparison with state-of-the-art solutions is missing, making it difficult to assess the novelty and competitiveness of the presented approach.

Detailed Comments:

1. Missing Performance Metrics:

Several key parameters of the structure are not reported, which prohibits a fair comparison with existing solutions:

- Response Time: What is the expected or measured response time of the LC device?
- Losses: Please provide data on the aperture efficiency of the system.
- Bandwidth: A 10% bandwidth is claimed, but the definition used (e.g., -3 dB, -10 dB,...) is not specified.

Include a section discussing these performance parameters. In addition, a comparative table listing these values alongside relevant state-of-the-art transmissive or reflective LC metasurfaces must be added to the main manuscript. The structure should also be compared to other thin LC designs available in literature. Other structures similar to these include: reflective metasurfaces, reconfigurable intelligent surfaces (RIS), reflectarrays, transmitarrays, and frequency selective surfaces.

2. Amplitude vs. Phase Control:

The choice of amplitude modulation instead of phase control is a central part of this work. This means that the beam steering is greatly suboptimal, since elements not contributing to certain points are not adapted to contribute, but just switched off. Although the approach is interesting, it requires a more detailed explanation:

- The fact that non-contributing elements are merely deactivated leads to suboptimal beamforming performance. This trade-off should be explicitly explained in the main text.
- The concept of aperture efficiency for different configurations (e.g., all-on, partially-on) should be quantified and discussed.
- The reason behind using a binary (on-off) control rather than continuous LC tuning is unclear and needs to be justified, since LC is capable of continuous tuning.

3. Control Scheme Explanation:

The authors mention a line-matrix control scheme, but it is unclear why this method is advantageous over an active matrix approach (as in standard LCD displays). Why is pixel-level control via Thin Film Transistors (TFTs) not feasible or practical in this context?

4. Formal and Presentation Issues:

- The manuscript contains numerous grammatical and stylistic issues. A thorough revision of the English language is recommended.
- The font size in figures and plots is too small and should be increased to match the main text font for readability.
- The introduction has no headline.
- There are several formal issues such as the referencing of figures. Please revise.

Taking these comments into account, I believe the authors should revise these comments before the manuscript can be accepted.

Reviewer #4

(Remarks to the Author)

I co-reviewed this manuscript with one of the reviewers who provided the listed reports. This is part of a Communications Engineering initiative to facilitate training in peer review and to provide appropriate recognition for Early Career Researchers who co-review manuscripts.

Version 1:

Reviewer comments:

Reviewer #1

(Remarks to the Author)

The authors explain the concerns, providing detailed reasoning, labeling the corresponding revisions, and adding supporting references. The improved manuscript quality enables readers to understand the technical details and underlying mechanisms more clearly. The work being reported is of good quality, with essential materials displayed to demonstrate its functionalities. I recommend publication as is.

Reviewer #2

(Remarks to the Author)

Summary, scope, and principal claims

The paper reports a transmissive reconfigurable intelligent surface (RIS) at 115 GHz based on a single-layer, 3.5 μm liquid-crystal (LC) gap and a stacked split-ring resonator (S-SRR) unit cell operated in a circulating-current mode. The demonstrator comprises 218 \times 218 cells in a 70 mm aperture (cell pitch $< \lambda/8$) addressed by line-matrix control with 109 row + 109 column channels, enabling binary amplitude modulation for 2-D collimation and beam steering from 0° to 30° in 5° steps. The authors measure an on/off modulation depth \approx 12 dB, near-field maps, and derive far-field patterns from a channel model; they provide design files on request. Key device and method details are consolidated in the revised manuscript and Supplementary Information.

Novelty and relation to prior work

Prior LC-based THz metasurfaces and RISs have typically used hundreds-of-micrometres LC thickness, often in reflective mode or with 1-D steering. A representative transmissive 1-bit LC coding metasurface (AOM 2021) showed programmable THz beam manipulation but with much thicker LC and smaller arrays. In parallel, a Communications Engineering 2024 study achieved sub-100 ms LC-RIS response with compact delay lines, but at 62 GHz and in reflection, not transmission; it is a useful architectural comparator. Related THz programmable metasurface work has explored crossbar addressing and 2-D steering, mainly reflectively. An Optics Letters 2022 paper also discusses LC-programmable THz metasurfaces. Against this backdrop, the LCD-grade 3.5 μm LC gap at 115 GHz in transmission and the large-aperture, 2-D line-matrix control are the most distinctive aspects here. The manuscript's comparison table and LC-thickness benchmark underscore this positioning. Overall, the work should interest both metasurface and 6G/RIS communities.

Technical soundness

Unit-cell physics and materials. The rebuttal now gives a clearer physical rationale for abandoning the anti-symmetric (magnetic-dipole) bilayer resonance at thin gaps by quantifying Q-factor degradation under realistic metal conductivity and LC loss tangent, and it introduces a hybridization/equivalent-circuit view for the S-SRR. These additions materially improve rigor.

Bias lines and loss budget. The revised analysis identifies ITO bias lines ($60 \Omega/\square$) as the dominant non-ideal contributor at 115 GHz (~20% dissipated power at resonance; transmittance peak limited to ~17 dB with ITO), correctly highlighting that raising line impedance is the most effective path to higher modulation depth. The main text and Supplementary quantify power dissipation pathways and explicitly note the approximation in the channel model. Aperture formation and steering. The beam-forming strategy (binary amplitude patterns P0–P7) is coherently tied to the Fresnel-zone condition (all-on optimal when the RIS is smaller than the first Fresnel zone) and to the known 10% theoretical efficiency ceiling of amplitude-only holography; these points are stated and referenced. The measured near-field maps versus calculated reference fields are consistent with the intended steering behaviour across 0° – 30° , for both orthogonal linear polarisations.

Bandwidth and polarisation. The authors appropriately temper “broadband/polarization-independent” claims to ~10% fractional bandwidth around 115 GHz and dual linear polarisation, and they add simulations/measurements sweeping the incident polarisation from 0° to 90° .

Dynamics and stability. Because the 218-ch feasibility driver is not speed-optimised, the authors characterise LC orientation response optically on a $3.5 \mu\text{m}$ cell ($\approx 3 \text{ ms}$ rise at 25°C ; $\approx 50 \text{ ms}$ fall), and discuss UV/thermal stability and mitigation (UV blocking, thermal management). These additions address prior concerns, albeit indirectly with respect to THz modulation.

Adequacy of responses to reviewers

The point-by-point rebuttal is careful and largely convincing:

Resonance physics and linewidth/Q—addressed with new simulations, geometry tables, and an equivalent-circuit discussion. OK

Loss budget & thermal stability—quantified bias-line loss; qualitative but reasonable thermal/UV discussion. Partially. (thermal data still limited).

Near- to far-field validity—channel model derived (Eq. S14) with explicit caveats (no polarisation/loss). Partially; a direct far-field validation remains desirable.

Bandwidth/polarisation claims—language corrected; additional polarisation sweeps provided.

Driver and reproducibility—schematic of the switch matrix (MOSFET relays) added; layout files available; fabrication alignment sensitivity and a two-sample comparison discussed.

Overall, the authors have substantially strengthened the manuscript; the remaining gaps are mainly experimental breadth (see below).

Statistical analysis and reproducibility

The work is primarily demonstration-driven; statistical treatment is minimal. There are no error bars on S-parameters or near-field magnitudes, no repeated-sample statistics beyond the alignment illustration, and no quantified uncertainties from probe positioning or calibration. The channel model is clearly stated, but its simplifications are acknowledged. Reproducibility improved with complete geometry (Supp. Sec. 12), driver schematic, and data/layout availability, which is commendable.

Recommendation: include uncertainty estimates (repeat scans; calibration repeatability; sensitivity to V_p) and, if feasible, a single far-field measurement on a scaled-down aperture or with a compact range to validate the model-based far-field claims.

Literature coverage

The revised text now references dielectric BIC metasurfaces as a promising low-loss path, aligning with reviewers' suggestions. Two important, still-missing comparators are recommended for citation: (i) transmissive LC 1-bit coding metasurface at THz frequencies (AOM 2021), which contextualises transmissive, digitally addressed LC devices, and (ii) Comms Eng. 2024 sub-100 ms LC-RIS with compact delay lines, which is reflective and lower-frequency but directly relevant to the system-level control architecture and response-time targets. For completeness on crossbar/2-D addressing at THz, citing recent Science Advances 2023 on modulo-addition steering would strengthen the discussion of discrete optimisation and line-matrix trade-offs.

Weaknesses and suggested experiments/analyses

Dynamic THz response. The optical ms-scale response is informative but should be complemented by THz-band electrical switching (e.g., rise/fall of the on/off ratio at 115 GHz under the revised driver).

Far-field validation. Provide at least one direct far-field data set (reduced aperture or compact-range) to benchmark the S14-based conversion.

Loss definition consistency. The claim of low “insertion loss” at the array scale should be reconciled with per-cell S21 (≈ -10 to -14 dB) to avoid ambiguity between transmission magnitude and holographic aperture efficiency. Clarify the metric used in the comparison table.

Polarisation and oblique incidence. The new polarisation sweep is useful; adding oblique-incidence data (even simulated) would frame window-mounted deployments more realistically.

Reliability. Consider including accelerated UV/thermal stress results or supplier specifications for DHB-012 to substantiate the stability discussion.

Editorial/minor

The authors addressed numerous clarity issues (terminology, scale bars, bandwidth phrasing), and the Data availability statement now explicitly mentions DXF layout provision—good.

Technically convincing with important caveats. The manuscript demonstrates a credible path to LCD-compatible, thin-gap LC transmissive RIS at 115 GHz with 2-D beam control and provides significantly improved physical and methodological grounding after revision. Addressing the THz-band switching dynamics, a minimal far-field validation, and metric consistency for “insertion loss” would remove the remaining doubts. I recommend acceptance after minor–moderate revision, contingent on clarifications/experiments as outlined above.

Reviewer #3

(Remarks to the Author)

Thank you for the thorough revision of the paper. We believe that our previous comments have been well addressed and that the technical quality of the manuscript has improved substantially.

Before accepting the paper, however, we find that the abstract remains superficial and should be improved to give the reader a clearer impression of the paper’s content. In particular, the following aspects should be mentioned more explicitly:

1 - The use of amplitude modulation

2 - The key performance metrics, i.e., bandwidth, insertion loss, scanning capabilities, cell size, and polarization. Please provide quantitative data of these metrics already in the abstract.

Reviewer #4

(Remarks to the Author)

I co-reviewed this manuscript with one of the reviewers who provided the listed reports. This is part of the Communications Engineering initiative to facilitate training in peer review and to provide appropriate recognition for Early Career Researchers who co-review manuscripts.

Reviewer #5

(Remarks to the Author)

I co-reviewed this manuscript with one of the reviewers who provided the listed reports. This is part of the Communications Engineering initiative to facilitate training in peer review and to provide appropriate recognition for Early Career Researchers who co-review manuscripts.

Version 2:

Reviewer comments:

Reviewer #2

(Remarks to the Author)

All my comments were properly addressed. I suggest acceptance in the current form.

Response to Reviewers

Manuscript ID: COMMS-ENG-25-0201-T

“Transmissive metasurface with 3.5- μm -thick liquid crystals for subterahertz-wave dynamic beamforming”

We thank the reviewers for their careful reading of our manuscript and their constructive criticisms and suggestions. Their comments have surely helped us improve the quality of the manuscript. Below you can find the detailed responses to the issues raised and an outline of our changes to improve the manuscript, with the authors' responses in blue. We have also included a revised version of the manuscript and supplemental material with changes highlighted in red.

Reviewer #1:

In this paper, the authors report a split-ring transmissive LC metasurfaces whose LCD-compatible 3.5- μm LC thickness can be determined independently of the scaling of the unit cell and can be redesigned over a broad frequency range from microwave to subterahertz waves. Through 12dB modulation depth of intensity, a prototype shows two-dimensional beam steering and focusing of linearly polarized waves in the 115-GHz band, having a bandwidth of 10% and polarization-independent characteristics. Line-by-line matrix control is used to simplify control. The work being reported is of good quality, with essential materials displayed to demonstrate its functionalities. To help readers understand, here's a list of specific points the authors should address:

Authors' response: The authors would like to thank the reviewer for the time, feedback, and constructive criticism. The comments by the reviewer are addressed below with all the diligence and effort to enhance the content of the manuscript.

Comment #1-1:

• *Technical concerns:*

1. In line 54, the authors write “To reduce the total propagation loss of the path through the metasurface to a level commensurate with the free space propagation loss (FSPL) of the total propagation path length, it is necessary that the length of one side of the metasurface be greater than or equal to the radius of the 1st Fresnel zone.” Please add references and more reasoning to this comment.

Authors' response #1-1: We sincerely appreciate your valuable comments on this important matter. We also apologize for the lack of clarity in addressing this point in the original manuscript. A detailed mathematical proof is provided below.

Let d denote the distance between a transmitter and a receiver. If there is a direct path, the corresponding free space propagation loss (FSPL) is given by

$$P_{\text{direct}} = \left(\frac{4\pi d}{\lambda} \right)^2, \quad (R1)$$

where λ is the wavelength.

Next, we consider the link via RIS with its aperture A . Let $d_{\text{Tx-RIS}}$ denote the distance between the Tx and RIS and $d_{\text{RIS-Rx}}$ the distance between RIS and Rx. The array gain of the antenna with aperture A is expressed as

$$G = \frac{4\pi A}{\lambda^2}. \quad (R2)$$

When the RIS with an array gain corresponding to its effective aperture area of $A \cos \phi_{\text{Tx-RIS}}$ receives arriving waves, the pathloss between the Tx and RIS is given by

$$P_{\text{Tx-RIS}} = \left(\frac{4\pi d_{\text{Tx-RIS}}}{\lambda} \right)^2 \cdot \frac{1}{G} = \frac{4^2 \pi^2 d_{\text{Tx-RIS}}^2}{\lambda^2} \cdot \frac{\lambda^2}{4\pi A \cos \phi_{\text{Tx-RIS}}} = \frac{4\pi d_{\text{Tx-RIS}}^2}{A \cos \phi_{\text{Tx-RIS}}}, \quad (R3)$$

where $\phi_{\text{Tx-RIS}}$ represents the incident angles on the RIS plane.

Similarly, the pathloss between RIS and Rx is given by

$$P_{\text{RIS-Rx}} = \left(\frac{4\pi d_{\text{RIS-Rx}}}{\lambda} \right)^2 \cdot \frac{1}{G} = \frac{4^2 \pi^2 d_{\text{RIS-Rx}}^2}{\lambda^2} \cdot \frac{\lambda^2}{4\pi A \cos \phi_{\text{RIS-Rx}}} = \frac{4\pi d_{\text{RIS-Rx}}^2}{A \cos \phi_{\text{RIS-Rx}}}. \quad (R4)$$

where $\phi_{\text{RIS-Rx}}$ represents the transmission angles on the RIS plane.

From (R3) and (R4), the total pathloss of the link via the RIS is expressed by

$$P_{\text{RIS}} = P_{\text{Tx-RIS}} P_{\text{RIS-Rx}} = \frac{4\pi d_{\text{Tx-RIS}}^2}{A \cos \phi_{\text{Tx-RIS}}} \cdot \frac{4\pi d_{\text{RIS-Rx}}^2}{A \cos \phi_{\text{RIS-Rx}}} = \left(\frac{4\pi d_{\text{Tx-RIS}} d_{\text{RIS-Rx}}}{A \cos \phi_{\text{Tx-RIS}} \cos \phi_{\text{RIS-Rx}}} \right)^2. \quad (R5)$$

In order to satisfy the condition that the total propagation loss of the path through the RIS is a level commensurate with the FSPL of the total propagation path length, the following inequality must hold:

$$P_{\text{direct}} \geq P_{\text{RIS}}, \quad s. t. \quad d = d_{\text{Tx-RIS}} + d_{\text{RIS-Rx}}. \quad (R6)$$

By comparing (R1) and (R5) with $d = d_{\text{Tx-RIS}} + d_{\text{RIS-Rx}}$, we find that

$$\left(\frac{4\pi d}{\lambda} \right)^2 \geq \left(\frac{4\pi d_{\text{Tx-RIS}} d_{\text{RIS-Rx}}}{A \cos \phi_{\text{Tx-RIS}} \cos \phi_{\text{RIS-Rx}}} \right)^2$$

$$A \geq \frac{\lambda d_{\text{Tx-RIS}} d_{\text{RIS-Rx}}}{d_{\text{Tx-RIS}} + d_{\text{RIS-Rx}}} \cdot \frac{\frac{d^2}{\lambda^2} \geq \frac{d_{\text{Tx-RIS}}^2 d_{\text{RIS-Rx}}^2}{A^2 \cos^2 \phi_{\text{Tx-RIS}} \cos^2 \phi_{\text{RIS-Rx}}}}{1} \geq \frac{\lambda d_{\text{Tx-RIS}} d_{\text{RIS-Rx}}}{d_{\text{Tx-RIS}} + d_{\text{RIS-Rx}}} \triangleq r_{\text{Fresnel}}^2, \quad (\text{R7})$$

where r_{Fresnel} is the Fresnel radius.

Assuming that the RIS is a square, the length of one side is given by \sqrt{A} . Therefore, from (R7), the required one side length of the RIS is the radius of the 1st Fresnel zone, i.e., $\sqrt{A} \geq r_{\text{Fresnel}}$.

With respect to this reviewer's comment, we have revised corresponding text in the manuscript and have included above explanation and related reference in revised supplemental material.

Corresponding revisions

in the revised manuscript: line 50 – line 54

in the revised supplemental material: Added a new section (Section 1) and reference [R1]

[R1] Tang, W. et al. Wireless Communications with Reconfigurable Intelligent Surface: Path Loss Modeling and Experimental Measurement. IEEE Trans. Wirel. Commun. 20, 421–439 (2021).

Comment #1-2:

2. In line 67, the authors write “Here, the response time has to be on the order of milliseconds for the LC metasurface to track a moving receiver”, please provide a reasoning for this statement as the response time is very scenario dependent.

Authors' response #1-2: We thank the reviewer for this important comment. We agree that the required response time is scenario dependent. The relationship between the half-power beam width (HPBW) and the aperture size can be expressed by

$$\frac{4\pi A}{\lambda^2} = \frac{4\pi}{\theta_1 \theta_2}, \quad (\text{R8})$$

where θ_1 and θ_2 are the HPBW in one plane and a plane at a right angle to the other, respectively [R2]. For example, for an RIS aperture size of 20λ square, the estimated HPBW is approximately 3 degrees. User equipment (UE) moving at walking speed (~ 1.3 m/s) would pass through this HPBW in about 1 second at a distance of 20 m away from the RIS. In scenarios where the UE moves faster, or the distance between the UE and RIS is closer, or the RIS aperture is larger than this assumption, a response time of millisecond order is desirable in order to establish a stable

wireless link. As the reviewer pointed out, the text in the original manuscript could be misinterpreted as requiring a millisecond response time regardless of the scenario. We have therefore revised the description.

Corresponding revisions in the revised manuscript: line 64 – line 67

[R2] Constantine A. Balanis. ANTENNA THEORY 4th Edition. (Wiley, 2016).

Comment #1-3:

3. In line 133, the authors write “concentrated in the LC layer without MD confinement (Fig. 2d)”, the E -field in Figure 2a asymmetric mode and Figure 2d seem alike, is there also an MD confinement in Figure 2d? What is the difference between these two modes in compressing the LC thickness?

Authors’ response #1-3: Thank you for your important question. The conceptual diagram of the cross-sectional E -field in Fig. 2d of the original manuscript was presented with a ratio that was not aligned with the actual relative positional relationship, potentially misleading the reader into perceiving the circulating-current mode of the S-SRR generates MD similar to the anti-symmetric mode of the bilayer cross dipole. In the circulating-current mode of S-SRR, from the cross-sectional view, no looped currents are excited that would produce MD; however, ED is produced (Fig. R1a). The electrical and magnetic resonance modes can be distinguished by characterizing the surface admittance, Y_s , and impedance, Z_s , as follows

$$Z_s = \frac{2\eta(1 - T + R)}{1 + T - R}, \quad (R9)$$

$$Y_s = \frac{2(1 - T - R)}{\eta(1 + T + R)}, \quad (R10)$$

where R is the reflection coefficient, T is the transmission coefficient, and $\eta = \sqrt{\mu/\epsilon}$ is the wave impedance of free space [R3]. Figure R1b shows the real part of the simulated Y_s and Z_s derived from equations (R9) and (R10) for the circulating-current mode of the S-SRR.

Fig. R1: (a) Conceptual diagram of the cross-sectional E-field and (b) simulated surface admittance and impedance for the circulating-current mode of the S-SRR.

The peak value of Y_s corresponding to ED is at around 115 GHz, where the circulating-current mode of the S-SRR is excited, while there is no Z_s peak corresponding to MD. This indicates that ED is excited in the S-SRR, unlike MD in the anti-symmetric mode of the bilayer cross dipole shown in section 2 of the original supplementary material.

In response to the reviewer’s comment, we have revised Fig. 2d and the text to clarify that the circulating-current mode excites the ED and added a supplemental section about this issue to the revised supplemental material.

Corresponding revisions

in the revised manuscript: line 141 – line 143, Figure 2d

in the revised supplemental material: Added a new section (Section 3)

[R3] Pfeiffer, C. & Grbic, A. Metamaterial Huygens’ surfaces: Tailoring wave fronts with reflectionless sheets. *Phys. Rev. Lett.* **110**, 197401 (2013).

Comment #1-4:

4. In line 180, the authors write “Above the V_p ($= 4 V$), the peak shift reached a saturation point”, and in line 186, “the states in which $V_p = 0$ and $8 V$ are applied to the S-SRR are referred to as the off and on states”, why $V_p = 8V$ is needed if the response is saturated at $4V$?

Authors’ response #1-4: Thank you for pointing this out. The statement “the peak shift reached saturation point at $4V$ ” was technically incorrect. The dependence of the resonant frequency and

on/off ratio on V_p is shown in Figs. R2(a) and R2(b). Although both the change in resonant frequency and on/off ratio characteristics become small when V_p exceeds 4 V, a slight change remains. Therefore, in the experiment, the on-state of the fabricated LC metasurface was set at 8 V, where the characteristic change with V_p is almost unobservable, to allow for a margin.

In response to reviewer’s comment, we have revised the statement “*The shift in the resonant peak was small when V_p was 4 V or higher, and at 8 V, no change could be observed.*” in the revised manuscript.

Fig. R2: Dependence of (a) the resonant frequency and (b) on/off ratio of the fabricated S-SRRs on V_p measured for x- and y- polarized incident waves.

Corresponding revision in the revised manuscript: line 201 – line 204

Comment #1-5:

5. In line 191, “control method was a line-by-line matrix control”, is it beneficial to perform greyscale instead of binary control? Is it possible to perform grey-scale line-by-line matrix control?

Authors’ response #1-5: Thank you for raising this insightful question. As the reviewer noted, it is possible to perform greyscale control for line-by-line matrix control by making the applied voltage between the top and bottom metal multibit instead of binary, and the implementation of the greyscale control could offer a benefit in the formation of transmissive waves.

In phase-modulation type lenses, the deviation of their quantized phase distribution from the ideal continuous phase distribution is reduced by the implementation of the greyscale control, thereby increasing the received power at the desired position [R4]. On the other hand, in the dynamic diffractive lens with amplitude modulation presented in this manuscript, the cell-state is determined to be transparent or opaque based on the basis of whether the waves transmitted

through each cell constructively interfere at the desired point. Therefore, considering only the received power at the designed receiving point, there is no benefit from greyscale control, so we decided to perform binary amplitude modulation. However, given the correlation between the intensity profile in the aperture plane and the transmissive beam pattern including sidelobes, the implementation of a greyscale control could have an effect such as suppressing sidelobes. We believe that multibit control should be considered, for example, in scenarios where interference is a problem. We have added a discussion of this important possibility of the greyscale control to the discussion part.

Corresponding revision in the revised manuscript:

line 296 – line 313, Added a new reference [R4]

[R4] J. W. Goodman. Introduction to Fourier Optics 4th Edition. (W.H. Freeman & Company, 2017).

Comment #1-6:

6. In line 243, the authors write “In the calculations, each cell was assumed to have an ideal modulation that was lossless in the on-state and completely non-transparent in the off-state, while the modulation depth in the measurements was finite at approximately 12 dB”, is it possible to incorporate the 12 dB modulation depth in the calculation to confirm the expected experimental performance?

Authors’ response #1-6: Thank you for this valuable suggestion. We have recalculated the beam profiles and near-field distribution with a finite 12-dB modulation depth and found a degradation of ~2 dB with respect to the ideal modulation for a modulation depth of 12 dB (Fig. R3a). Similarly, we found a degradation in the focus gain of the focusing function (Fig. R3b). Although there is a discrepancy of 1.5 dB between the measured and the recalculated focus gain, we believe that it can be narrowed by considering the effect of the finite aperture of the probe antenna used in the experiment and the measurement error due to unwanted reflections in the experimental system. To increase modulation depth and improve the received power at the desired position, it is effective to suppress the power dissipated in the bias line, as discussed in the Authors' response #2-3. This issue will be addressed in future work.

According to the reviewer's suggestion, all the calculated results of beam profiles and near-field distributions in the original manuscript have been replaced with recalculated results incorporating a finite modulation depth of 12 dB based on the measurements. In addition, a discussion on the validity of the experimental results has been added.

Fig. R3: (a) Calculated far-field patterns for control patterns P0 to P7 with 12 dB or ideally infinite modulation depth. (b) Measured and calculated focus gain on the central axis for control pattern P8.

Corresponding revision in the revised manuscript:

Line 242 – line 243, line 280 – line 292, Revised figures (Figures 5b, 5d, 6a, and 6c), Added a new figure (Fig. 6d)

Comment #1-7:

7. In line 384, the authors write “a 100-nm-thick chromium layer, and a 500-nm-thick copper layer”, the metals are thick, is it possible to further reduce the metal layer thickness?

Authors’ response #1-7: Thank you for this question. Yes, it is possible to make the metal layers thinner. In the proposed S-SRR, the variable capacitance component, which plays an important role in controlling resonance characteristics, is determined by the distance between the top and bottom metals and the overlap area. A reduction in the thickness of the metal layers increases the distance between the top and bottom metals, resulting in a decrease in the capacitance component. Therefore, it is hypothesized that by making adjustments such as

- (i) increasing the overlap area to increase the capacitance component, or
- (ii) lengthening the ring part of the S-SRR to increase the inductance component,

similar scattering characteristics can be obtained with reduced metal thickness and a display-grade LC thickness.

Comment #1-8:

8. *In line 427, in Figure 2f-h, the overlapping area is redesigned for each frequency band, so the unit cell transparency is also related to the operational frequency band, is there a way to further increase the transparency of the device?*

Authors' response #1-8: We appreciate this important question. As the reviewer pointed out, the overlap area, cell size, and line width of the metal pattern vary with operational frequency band, thereby affecting the optical transparency of the RIS. The optical transparency of the RIS is associated with the landscape impact of device deployment; therefore, enhancing the transparency is very important from a practical application perspective. To enhance the transparency, the following two approaches are currently under consideration.

One is to use low-resistance ITO for the S-SRR pattern and high-resistance ITO for the control bias lines. Given the higher transparency of the low-resistance ITO in comparison to the copper used in this manuscript, an enhancement of the RIS transparency can be expected as well. However, it should be noted that this approach increases the sheet resistance of the S-SRR pattern, resulting in a trade-off between transparency and modulation depth.

The other way is to mesh the metal pattern excluding the overlap area, as is done in [R5]. Since the overlap area constitutes a small percentage of the current metal pattern area, meshing the areas other than the overlap part is expected to significantly enhance the transparency.

In response to the reviewer's important comment, we have added a discussion about the optical transparency of the device to the revised manuscript.

Corresponding revision in the revised manuscript: L371 – L385

[R5] Kitayama, D. et al. Transparent dynamic metasurface for a visually unaffected reconfigurable intelligent surface: controlling transmission/reflection and making a window into an RF lens. *Opt. Express* 29, 29292–29307 (2021).

Comment #1-9:

9. *In lines 466 and 478, in Figures 5 and 6, what is the maximum and minimum steering efficiency among the scanned angles? Why steering gain or efficiency is low at large angles, and what are the limited factors?*

Authors' response #1-9: Thank you for raising this important question. Theoretically, the maximum aperture efficiency for binary wavefronts is 41% for phase modulation and 10% for amplitude modulation [R6]. According to the gain of the collimated transmissive waves for 12-dB modulation depth and for the ideal modulation depth shown in the Authors' response #1-6, the efficiency of the proposed structure with amplitude modulation can be estimated to be

approximately 2 dB lower than the theoretical value. When the modulation depth is greater than 20 dB, the efficiency reduction relative to the ideal modulation is less than 1 dB (see Authors' response #2-5). Hence, we intend to increase the modulation depth of the proposed structure in the future.

As for the decrease in steering efficiency at large angles, we believe the following factors attribute to it.

- As the angle increases, the effective aperture area decreases; in other words, the directivity of the scattered waves for each cell is smaller at larger angles.
- For larger angles, the spatial frequency of amplitude modulation is increased, and the error between the ideal profile and the one formed by an RIS with the finite control unit size becomes large. For instance, in the binary pattern for a 30-degree deflection of one-dimensionally collimated waves, the binary pattern with a resolution of 0.64 mm has a total error width of 8.2 mm relative to the ideal binary profile, which is 1.65 times wider than that for 0-degree deflection. (Fig. R4).

The proposed S-SRR structure can be made to have a small cell size by using circulating-current mode and is expected to suppress the steering inefficiency at large angles more than other structures.

In addition to the aforementioned factors, inadequate optimization of wavefront control may also compromise steering efficiency at large angles. When quantizing wavefronts as in binary modulation, finding the optimized modulation profile is a discrete optimization problem. This makes the optimization of modulation profile for each angle an NP-hard problem [R7]. For 218 binary controlled channels as in the manuscript, there are 2^{218} combinations of control pattern. In the manuscript, the binary profile of control channels for each row and column is determined based on the position exhibiting the least phase change in the analytically obtained continuous wavefront, which is not guaranteed to be the optimum discrete control. Although the main purpose of this study is to demonstrate a cell structure that can control scattering properties even with LCD-grade thin LC layers, it is also important to investigate discrete optimization methods of control to maximize RIS performance; such studies are ongoing in our group and will be reported in a separate paper. In response to the reviewer's comment, we have included a discussion on the estimation of maximum efficiency and the challenges of optimizing binary modulation in the main text. We have also added relevant references.

Fig. R4: Diagram showing degree of discrepancy between the formed and ideal binary profile depending on the cell size and the steering angle.

Corresponding revisions

in the revised manuscript: line 246 – line 256, Added a new references [R6, R7]

in the revised supplemental material: Added a new section (Section 9)

[R6] Oggioni, L., Pariani, G., Zamkotsian, F., Bertarelli, C. & Bianco, A. Holography with Photochromic Diarylethenes. *Materials* 12, 2810 (2019).

[R7] Shtaiwi, E. et al. Sum-Rate Maximization for RIS-Assisted Integrated Sensing and Communication Systems With Manifold Optimization. *IEEE Trans. Commun.* 71, 4909–4923 (2023).

Comment #1-10:

• *Description concerns:*

1. In line 36, the authors write “usage of the finite frequency resource”, as sub-THz also contains finite resources, this description is inaccurate.

Authors’ response #1-10: Thank you for pointing out this. We agree that the description in the original manuscript was not accurate. We have revised it to “*the currently used frequency resource*”.

Corresponding revision in the revised manuscript: line32 –line33

Comment #1-11:

2. In line 64, the authors write “is parallel or perpendicular to the local electric (E -) field excited in the metasurface.” The excited E -field may refer either to the E -field in the metasurface mode

excited by the THz wave or the E-field generated by electric control, which will confuse. This description appears several times in the article, please check and confirm.

Authors' response #1-11: We thank the reviewer for this insightful comment. The descriptions of “*the local electric field*” used in the manuscript mean the “*E-field in the metasurface resonant mode excited by the incident waves*”. We have revised such the descriptions in the main text.

Corresponding revision in the revised manuscript:

line 60 – line 61, line 89 – line 90, line 148 – line 149.

Comment #1-12:

3. In line 263, the authors write “works over a wide frequency range (from sub-6 GHz to over 100 GHz).” This statement might be misunderstood as the bandwidth of a single design, need more clarification.

Authors' response #1-12: We agree that the description in the original manuscript is misleading. We have revised it to “*In conclusion, we devised an S-SRR structure with a constant LC thickness that can be easily redesigned by scaling over a wide frequency range*”.

Corresponding revision in the revised manuscript:

line 22 – line 23, line 98 – line 100, line 395 – line 396

Comment #1-13:

4. In line 427, in Figure 2e, the scale bar of the E-field plot is from 0~1 and is inconvenient to compare with the E-field in Figure 2e, whose scale bar is from -1~1. Please unify the scale bar.

Authors' response #1-13: Thank you for this comment. To make the figures easy to compare, we unified the scale bar of figure 2a and 2e as you pointed out.

Corresponding revision in the revised manuscript: Figure 2a and 2e.

Comment #1-14:

5. In line 439, “f-g” should be “f-h”.

6. In line 445, in Figure 3, figure label e is missing.

Authors' response #1-14: We sincerely thank the reviewer for careful reading. We have corrected the text and the figure label you pointed out. The whole manuscript has been carefully proofread as best as we can.

Corresponding revision in the revised manuscript: line 594, Figure 3

Reviewer #2:

The manuscript entitled “Transmissive metasurface with 3.5 μm thick liquid crystals for sub terahertz wave dynamic beamforming” addresses an important bottleneck in 6 G wireless engineering by coupling a liquid crystal (LC) layer of display grade thickness with a transmissive reconfigurable intelligent surface (RIS) based on a stepped split ring resonator (S SRR). While the general concept is timely and potentially impactful, the theoretical framework and experimental validation leave critical questions unanswered.

Authors’ response: The authors greatly appreciate the reviewer’s insightful comments. We admit that the original manuscript did not provide sufficient results on the concerns of the reviewer. In this response, we tried to address all the comments from the reviewer as thoroughly as possible.

Comment #2-1:

First, the electromagnetic analysis rests almost entirely on full wave HFSS simulations; a complementary circuit level or coupled mode model is not provided, so the individual roles of capacitance, parasitic inductance and sheet resistance in determining the resonance quality factor and the observed 12 dB modulation depth remain opaque.

Authors’ response #2-1: The authors appreciate your perspective on these important matters.

For the coupled-mode model of the bilayer resonators, the hybridized resonant mode for generalized structures is modeled as an analogy of molecular orbital theory in [R8]. And the models of antisymmetric and symmetric modes, which are hybridized modes of the bilayer dipole resonators shown in Fig. 2a in the original manuscript, are well explained in [R9]. Figure R5 shows the hybridization scheme for the bilayer dipole resonators. In the antisymmetric mode, current in opposite directions are excited in each layer, which results in attractive forces in the system and the appearance of a low energy (or low frequency) mode. Conversely, in the symmetric mode, the currents in the layers flow in the same direction, which results in the repulsive forces within the system and a high energy (or high frequency) mode. As the LC layer becomes thinner, the interaction of the excited charges in the different layers strengthens. This results in a large separation of the two hybridized modes in frequency as shown in Fig. 2c in the original manuscript. We have added an explanation of hybridized resonant modes for bilayer dipole resonators modeled by analogy of molecular orbital theory to the main text. We have also added related references.

On the other hand, the circulating-current mode of the proposed S-SRRs is not a hybridized mode. The proposed S-SRR’s equivalent circuit for circulating-current mode is shown in Fig. R6a. The S-SRR is composed of a ring part, corresponding to the inductance, and a gap part, corresponding to the capacitance component. The resonant frequency depends on these LC values. The metal conductivity is a determining factor in the parasitic resistance of the S-SRR, which in turn affects the quality factor of the resonance. Furthermore, it is imperative to suppress the parasitic component of the variable capacitance to enhance the controllability of the resonance

characteristics by changing the LC orientation, which leads to a large modulation depth. Making the capacitance component perpendicular to the substrate, as opposed to its placement in the same plane as in conventional SRRs, greatly suppresses parasitic capacitance. The formation of a capacitance component across the LC layer perpendicular to the substrate allows the local electric field excited by the incident waves to concentrate on the LC layer, preventing the glass substrate with a constant permittivity from being a parasitic capacitance component (Fig. R6b).

Fig. R5: (a) Simulated transmittance for the bilayer dipole resonators (Fig. 2c in the original manuscript) and (b) the hybridization scheme of symmetric and anti-symmetric modes.

Fig. R6: (a) Schematic diagram of the S-SRR and its equivalent circuit for the circulating-current mode. (b) Schematic cross section image of the gap made parallel or perpendicular to the substrate.

To clarify the role of the material parameters in determining the modulation depth of the S-SRR, we have simulated the impact of the metal conductivity, loss tangent of the glass substrate, and loss tangent of the LC layer on the resonant peaks (Fig. R7). For the ideal material parameters (PEC for metal, glass loss tangent of 0, LC loss tangent of 0), the resonance peak of the S-SRRs is ~ 80 dB. When each material parameter of the conductivity or loss tangent are varied, it is found that the smaller the metal conductivity or the larger the loss tangent is, the smaller the resonant

peak becomes. For the parameters of the materials used in this study (metal conductivity of 4.65×10^7 S/m, glass loss tangent of 0.01, LC loss tangent of 0.014), the metal conductivity has the largest effect on the decrease in the resonant peak. However, as shown in the Authors' response #2-3, the existence of the bias line of $60 \Omega/\text{sq}$ reduces the resonant peak to ~ 17 dB, while all the peaks for the actual material parameters in Fig. R7 exceed 25 dB. This indicates that making the impedance of the bias line high is an effective way to improve the modulation depth of the S-SRR. Since enhancing the modulation depth leads to an increase in received power at the desired position, we plan to investigate the design of the bias lines as part of our future work.

In response to the reviewer's important comment, we have added an explanation about the hybridization scheme of the bilayer cross dipole and related references in the main text, and an equivalent circuit of the S-SRR as a Fig. 2f. In addition, we have included a discussion about the role of each material parameter in the main text and the revised supplementary material.

Fig. R7: Simulated transmittance for the S-SRR without the bias lines, in which (a) the metal conductivity, (b) loss tangent of glass substrate, or (c) loss tangent of LC layer is varied, with all the material parameters except for the changed ones being ideal.

Corresponding revisions

in the revised manuscript: line 113, line 116 – line 120, line 141 – line 145, line 164 – line 170, Added a new Figure (Fig. 2f) and new references [R8.R9]

in the revised supplemental material: Added a new section (Section 4)

[R8] Prodan, E., Radloff, C., Halas, N. J. & Nordlander, P. A Hybridization Model for the Plasmon Response of Complex Nanostructures. *Science* 302, 419–422 (2003).

[R9] Kanté, B., Burokur, S. N., Sellier, A., De Lustrac, A. & Lourtioz, J. M. Controlling plasmon hybridization for negative refraction metamaterials. *Phys. Rev. B* 79, 075121 (2009).

Comment #2-2:

The paper claims that the antisymmetric mode of a conventional bilayer resonator “collapses” when $t_{LC} < 30 \mu\text{m}$, yet this assertion is supported only qualitatively (Fig. 2c) without linewidth data, temperature sweeps or loss tangent measurements that would confirm the Q factor degradation and its physical origin.

Authors’ response #2-2: The authors thank the reviewer for this important point. The original manuscript lacked information about the dimension of the simulated bilayer resonator. The geometry and dimensions of the bilayer cross dipole, including the linewidth parameter, are shown in Fig. R8 and Table R1. Section 2 of the original supplemental material only discussed the effect of only the metal conductivity on the Q factor of the antisymmetric mode for the conventional bilayer resonator. Therefore, we have also simulated the effect of loss tangent of 30- μm -thick LC layer. As shown in section 2 of the original supplemental material, for the ideal material parameters, the Q factor for MD or $\text{Re}(\mathbf{Z}_s)$ is higher for a thinner LC layer; the Q factor of $\text{Re}(\mathbf{Z}_s)$ for a 30- μm -thick LC layer exceeds 20000. Figure R9 shows the simulation results for the change in Q factor when the metal conductivity and loss tangent of the LC layer are varied. When the metal conductivity is $5.8 \times 10^7 \text{ S/m}$, which is the conductivity of copper in the EM simulation, Q factor of $\text{Re}(\mathbf{Z}_s)$ decreases to ~ 100 . Similarly, when the loss tangent of the LC layer is 0.01, Q factor of $\text{Re}(\mathbf{Z}_s)$ decreases to ~ 180 . These results indicate that, when using the anti-symmetric mode with an LC thickness comparable to that of LCDs, both the metal conductivity and loss tangent of the constituent materials of the metasurface must be ideal to avoid the collapsed resonance peak, which we believe is a considerable challenge.

In response to the reviewer’s important comment, we have included the above discussion in section 2 of the revised supplemental material and added the geometry and dimension data including linewidth to section 12 of the revised supplemental material.

Fig. R8: Geometry and dimensions of the bilayer cross dipole.

Table R1 Geometric parameters of the bilayer cross dipole.

		P	A	W_{Cu1}	W_{Cu2}
Fig. 2a-c	Sim.	950 μm	770 μm	50 μm	50 μm

Fig. R9: Simulated Q factor of $\text{Re}(Z_s)$ for the bilayer cross dipole with 30- μm -thick LC layer.

(a) Dependence of real part of Z_s for the bilayer cross dipole with a lossless LC layer on metal conductivity. (b) Dependence on metal conductivity of Q-factor of Z_s for the bilayer cross dipole with a lossless LC layer. (c) Dependence of real part of Z_s for the PEC-patterned bilayer cross dipole on loss tangent of the LC layer. (d) Dependence on loss tangent of the LC layer of Q-factor of Z_s for the PEC-patterned bilayer cross dipole.

Corresponding revisions in the revised supplemental material: Section 2, Section 12

Comment #2-3:

Similarly, the mitigation strategy—substituting copper bias lines with ITO—relies on simulated S21 curves but lacks a quantitative assessment of power dissipated in the $60 \Omega/\square$ ITO layer or its thermal stability at 115 GHz.

Authors' response #2-3: The authors thank the reviewer for pointing this out. We have conducted additional simulations to further clarify the influence of the bias lines. Figure R10 shows the simulated power dissipated in the S-SRR ($1-|S_{21}|^2-|S_{11}|^2$) without the bias line. Here, the loss tangent or metal conductivity is varied, with all the material parameters except for the changed ones being ideal. The peaks of the dissipated power are at a higher frequency than the resonance peak of the transmittance (115 GHz), and for each material parameter used in this study (metal conductivity of 4.65×10^7 S/m, glass loss tangent of 0.01, LC loss tangent of 0.014), the dissipated power at 115 GHz is less than 10% in all cases. In the S-SRR, by incorporating the 5- μm width bias line, the resonance peak of the transmittance decreases and more power is dissipated as the sheet resistance of the bias line decreases (Fig. R11). For the sheet resistance of the ITO employed in this study ($60 \Omega/\text{sq}$), the resonance peak of the transmittance was reduced to 17 dB, and the dissipated power at 115 GHz was 20%, which is large compared with the results for other material

parameters. As outlined in the Authors' response #2-1, these results indicate that the most effective way to enhance the modulation depth is to suppress the dissipated power in the bias line.

Fig. R10: Dissipated power in the S-SRR ($1-|S_{21}|^2-|S_{11}|^2$) without the bias line. In this simulation, (a) the metal conductivity, (b) loss tangent of glass substrate, or (c) loss tangent of LC layer is varied, with all the material parameters except for the changed ones being ideal.

Fig. R11: (a) Simulated transmittance and (b) the power dissipated in the S-SRR ($1-|S_{21}|^2-|S_{11}|^2$) with the bias line, in which the sheet resistance of the bias lines is varied and the other material parameters are set to be ideal.

Although an investigation of the thermal stability of the LC RIS is beyond the scope of this study, the reviewer's perspective on thermal stability is crucial for practical applications. Therefore, we have discussed it with the LC material supplier. The LC RIS presented in this manuscript contains no active elements that generate heat; however, given the anticipated use case of an outdoors installation, temperature increases due to sunlight must be considered. The metal, glass substrate, and ITO for the bias line used in this study are all stable materials, even at

temperatures above 100°C. With regard to the LC material, one of the most important components of the LC RIS, the nematic-isotropic phase transition temperature of the used LC material (DHB-012) is approximately 160°C. However, it should be noted that the dielectric properties of LC change with temperature, even within the nematic phase. For instance, as the temperature rises, the response speed of the LC orientation change increases, while the disparity in the permittivity of ϵ_{\perp} and ϵ_{\parallel} ($\Delta\epsilon$) decreases. Although there was no experimental data in the radio-wave band, we have obtained data in the lightwave band at λ of 589 nm (Table R2). Even when the temperature rises from 25°C to 60°C, the rate of change in the permittivity ($\Delta\epsilon/\epsilon_{\perp}$), which is important for controlling the scattering characteristics of the S-SRR, only decreased by 5%. There is a similar difference in permittivity of the used LC material between the GHz-order radio-wave band and the lightwave band, approximately 10%, as shown in the Authors' response #2-7. It can be estimated that the degradation in the rate of change in permittivity within the range of 25°C to 60°C is a few percent in the radio-wave band, similar to that in the lightwave band. As the reviewer noted, to transition LC RIS technologies from the research phase to practical application, thermal stability and characteristics in the frequency band used for each use case must be evaluated, so we have included this issue in the discussion part. We also clarified that our experiments were conducted at room temperature.

Table R2 Temperature dependency of the permittivity of the LC material used in the manuscript measured at λ of 589 nm.

Temperature [°C]	25	30	40	50	60
ϵ_{\parallel}	3.76	3.74	3.71	3.67	3.63
ϵ_{\perp}	2.35	2.35	2.34	2.34	2.33
$\Delta\epsilon (\epsilon_{\parallel} - \epsilon_{\perp})$	1.41	1.40	1.36	1.33	1.29
$\Delta\epsilon/\epsilon_{\perp}$	0.60	0.60	0.58	0.57	0.55

Corresponding revisions

in the revised manuscript: line 197 – line 198, line 391 – line 394, Figure 3

in the revised supplemental material: Added a new section (Section 4)

Comment #2-4

On the experimental side, the near field maps collected over a 150 mm × 200 mm scan (Fig. 5e–f) are converted to far field patterns via equation (S6), yet that conversion assumes perfect periodicity and neglects polarization effects; direct far field data would be essential to validate the claimed two dimensional steering accuracy ($\pm 2^\circ$).

Authors' response #2-4: The authors thank the reviewer for pointing this out. We agree with the reviewer's comment that the simulation and experimental results presented in the original manuscript are insufficient to define the steering accuracy. Therefore, we have removed the claim about the steering accuracy. As the reviewer noted, the calculation based on equation S6 does not consider the effects of polarization. However, this calculation is based on a finite number (47524) and size (320- μm square) of cells like in the fabricated LC metasurface, and we believe that it is possible to investigate a methodology to obtain high steering accuracy. When a discrete wavefront is formed, such as in a metasurface, the beam control accuracy should be correlated with the wavefront resolution, i.e., the cell size. Figure R12 shows the calculated deflection angle error of the transmitted wave through the intensity profile of the metasurface with respect to the designed steering angle when the cell size is changed. The designed angle is set to from 0 to 5 degrees and 25 to 30 degrees in 1-degree steps, and the size of the metasurface is constant at 70 mm square regardless of the cell size. As the cell size increases, the wavefront resolution deteriorates, and the error in the deflection angle relative to the design angle increases. The error also tends to increase as the designed deflection angle increases.

Fig. R12: Calculated deflection angle of the transmitted waves through the intensity profile of the metasurface with respect to the designed steering angle and the cell size.

a Calculated far-field beam patterns for designed angle of from 0 degrees to 5 degrees and **b** 25 degrees to 30 degrees with respect to the cell size. **c** An error in deflection angle for designed angle of from 0 degrees to 5 degrees and **d** 25 degrees to 30 degrees with respect to the cell size.

This is believed to be due to the fact that the required resolution of the intensity profile becomes finer as the angle increases (see Authors' response #1-9). Although the unit cell size of the fabricated LC metasurface is 320- μm square, the resolution of the intensity profile is 640 μm since two rows or columns are bundled together into one channel. To improve the steering accuracy, the number of channels could be increased to enable control of each row and column, or the unit cell size could be reduced.

In response to the reviewer's comment, we have removed the claim regarding steering accuracy and have instead included a discussion of the correlation between steering accuracy and cell size in the main text and supplemental material.

Corresponding revisions

in the revised manuscript: line 247 – line 256

in the revised supplemental material: Added a new section (Section 9)

Comment #2-5

The reported 12 dB on/off contrast (Fig. 4d–e) is below the ≥ 20 dB generally required to compensate link budgets in indoor RIS deployments, and the authors do not quantify how ohmic losses in copper, dielectric loss in the LC or bias line impedance non uniformity each contribute to this limitation.

Authors' response #2-5: We thank the reviewer for this important comment. We have analyzed the effect of metal conductivity, the loss tangent of the LC and substrate, and the sheet resistance of the bias line on the modulation depth (see Authors' response #2-1 and #2-3). This indicated that the dissipated power in the bias line, which should be due to ohmic loss, is the main limiting factor of the modulation depth. Figure R13 shows the calculated results of the relationship between modulation depth and focus gain of the transmitted waves. The larger the modulation depth is, the higher the focus gain becomes at the designed place. When the modulation depth is set to 12 dB, the gain degradation is approximately 2 dB compared with the ideal infinite modulation depth. If the modulation depth is improved to 20 dB as noted by the reviewer, the gain degradation can be suppressed to less than 1 dB. This indicates that further optimization of impedance and design of the bias line would be effective in improving the gain. In addition to enhancing the modulation depth, there is also an approach to compensate for the gain by making a large aperture area of several tens of centimeters, which is the advantage of our RIS having an LCD level of LC thickness. Moreover, we will discuss with telecom operators their requirements for the cost and size of the RIS.

In response to the reviewer's comment, we have included a discussion about the correlation between the modulation depth and the gain in the main text and supplemental material.

Fig. R13: Measured and calculated focus gain on the central axis for control pattern of P8.

Designed focus is 100 mm away from the metasurface plane. The modulation depth in calculations are varied from 3 dB to ideally infinite value.

Corresponding revisions

in the revised manuscript: line 164 – line 170, line 179 – line 185, line 280 – line 292, Added a new figure (Fig. 6d)

in the revised supplemental material: Added a new section (Section 4)

Comment #2-6

Crucially, the dynamic performance is inferred rather than measured: no rise/fall times or long term stability tests are reported, yet millisecond scale switching—and robustness under continuous 1 kHz driving—are central to RIS operation in mobile scenarios. Statistical rigor is also limited: only one 70 mm square sample, containing 47 524 unit cells, is characterized, and no cell to cell variation, wafer to wafer reproducibility or measurement uncertainty is discussed.

Authors' response #2-6: We thank the reviewer for this important comment. The objective of this study is to demonstrate a unit cell structure that can make an LC layer as thin as an LCD, and the multi-channel switch used in the feasibility verification was not appropriate for an evaluation of the response time. The rise and fall times of the LC metasurface are determined by the response time of the LC when the control driver is constructed like an LCD driver. Therefore, we discussed with the LC material supplier and measured the response time of the LC orientation change. Figure R14 shows the measured transmittance of light, which varies with the LC orientation, for a LC

sample with an LC layer thickness of $3.5\ \mu\text{m}$ as is the case for the proposed structure. In this experiment, a 1-kHz square wave with a peak voltage (V_p) of 8 V was applied, and the temperature was varied from 25°C to 60°C . During the rise time when V_p changed from 0 V to 8 V, the response time became shorter as the temperature increased; it was approximately 3 ms at room temperature (25°C) and approximately 1 ms at 60°C . Similarly, during the fall time when V_p changed from 8 V to 0 V, the response time became shorter as the temperature increased: it was approximately 50 ms at 25°C and approximately 30 ms at 60°C . We have included this explanation about the response time in the revised manuscript and experimental results in the revised supplemental material.

Fig. R14: Measured transmittance of light for determining the response time of the LC orientation change.

a Rise time measured by varying V_p from 0 V to 8 V. **b** Fall time measured by varying V_p from 8 V to 0 V. The LC thickness of the sample, $3.5\ \mu\text{m}$, is the same as that of the LC metasurface presented in the manuscript.

The long-term stability mentioned by the reviewer is an important point for practical applications, and we have discussed this issue with the LC material supplier. When a RIS is installed on a window as an indoor/outdoor interface as shown in Fig. 1 of the original manuscript, measures may be required to prevent the LC layer, which has an absorption spectrum in the ultraviolet (UV) band, from being exposed to UV. Specifically, the long-term stability of the RIS could be improved by employing UV protective glass or film, as in the case with outdoor LCDs used for digital signage. In addition, heat can damage LC. Although RIS does not have an issue with self-emitted heat from a backlight and its thermal environment is relatively less severe than that of outdoor LCDs, the implementation of passive or active cooling systems should be nonetheless considered. We have included a discussion about the long-term stability and possible solutions regarding the effects of UV and heating in the main text.

As the reviewer noted, the reproducibility from a manufacturing perspective is a critical factor in transitioning LC metasurfaces from the prototyping phase to the mass production phase. In contrast to semiconductor devices, it was not possible to obtain multiple LC RIS from a single

mother glass during the prototyping process. Therefore, we do not have enough data to discuss the statistics of fabrication reproducibility currently. However, we believe that at least the alignment accuracy between the top and bottom metal is an important manufacturing parameter. Figure R15 shows photos of two fabricated samples and their transmission characteristic, in which sample A that is discussed in the main text exhibits a small alignment error of $\sim 4 \mu\text{m}$ and sample B exhibits a large alignment error of $\sim 15 \mu\text{m}$ in the y direction. When an error occurs in the alignment, the overlap area between the top and bottom metal decreases, reducing the capacitance component of the S-SRR. This causes the resonance frequency to shift to a higher frequency than the designed one, consequently resulting in poor manufacturing yields. LCD manufacturers have achieved high-precision alignment (error of $1 \mu\text{m}$ or less) with repeatability. The proposed structure, which can be fabricated using the same process as LCDs, is thus considered to exhibit sufficient reproducibility in the mass-production process. As reference to fabrication reproducibility, we have incorporated the discussion about the importance of alignment between the top and bottom substrates in the supplemental material.

Fig. R15: Photos of two fabricated samples and their transmission characteristic.

Sample A with a small alignment error of $\sim 4 \mu\text{m}$ is discussed in the manuscript. Sample B exhibits a large alignment error of $\sim 15 \mu\text{m}$ in the y direction.

Corresponding revisions

in the revised manuscript: line 197 – line 198, line 316 – line 319, line 386 – line 391, line 542 – line 544

in the revised supplemental material: Added new sections (Section 11 and 14)

Comment #2-7

Reproducibility would be greatly enhanced by releasing the GDS layout, full LC permittivity dispersion curves and the driver electronics schematic.

Authors' response #2-7: Thank you for your suggestion. Although the detailed dimensions of the proposed structure are described in section 6 of the original supplemental material, we agree with the reviewer that the layout data of each layer may help the readers follow our study. We have specified in the Data availability statement that the layout files are available from the corresponding author.

We also agree that detailed permittivity information will enhance reproducibility if readers use the same LC material (DHB-012) as ours. Although we were unable to get the dispersion curve information, the LC permittivity and loss tangent data at 10 GHz and λ of 589 nm (Table R3) was disclosed by the LC material supplier (DIC Corp., Japan). The changes in $\epsilon_{//}$, ϵ_{\perp} , and tunability from 10 GHz to the lightwave band are approximately 2%, 10%, and 7%, respectively, suggesting that nearly constant dielectric properties can be obtained within the range of several percent of the relative bandwidth typically necessary for mobile communication systems. Given the superiority of continuous dispersion curves in guiding the design of the LC metasurfaces within the desired frequency band, we will continue to have discussions with the LC material supplier regarding the measurement of dispersion curves over a broad frequency range. We have added table R3 to the revised supplemental material.

Table R3 Permittivity and loss tangent data measured at 10 GHz and λ of 589 nm.

	DHB-012 (DIC corp.)	
	10 GHz	589 nm
$\epsilon_{//}$	3.84	3.76
ϵ_{\perp}	2.65	2.35
Tunability ($\epsilon_{//} - \epsilon_{\perp}$)/ $\epsilon_{//}$	30.9%	37.6%
$\tan\delta_{//}$	0.0063	-
$\tan\delta_{\perp}$	0.0144	-
Transition temperature	160	

Regarding the reproducibility of the 218-ch switching circuit used in the feasibility study of the proposed structure, we added schematic information (Fig. R16) of the switching part to Fig. S4 in the original supplemental material. The switch for connecting to the signal or ground in each channel is composed of two MOSFET (metal-oxide-semiconductor field-effect transistor)-based semiconductor relays, to be precise.

Fig. R16: Schematic connections between fabricated LC metasurface and the 218-channel switch used in this study.

Corresponding revisions

in the revised manuscript: line 566 – line 570

in the revised supplemental material: Section 7, Added a new section (Section 13)

Comment #2-8

From a broader perspective, the authors position their contribution as “broadband” and “polarization independent,” yet evidence is confined to a 10 % fractional bandwidth around 115 GHz and to two orthogonal linear polarizations; oblique and circular states are not tested, nor are frequencies below 100 GHz where LC loss rises sharply.

Authors’ response #2-8: The authors thank the reviewer for pointing this out. To interpolate the results from the two orthogonal linear polarizations, we performed simulations and measurements by changing the polarization of the incident wave from 0 to 90 degrees in 15 degree steps (Fig. R17). As described in the original manuscript, the difference in the ITO length resulted in variations in the resonance peak intensity at 0 degrees (x polarization) and 90 degrees (y polarization). The additional simulations and measurements serve to interpolate between these values. We have incorporated these results into the supplemental material. As pointed out by the reviewer, the term “polarization independent” is misleading in that it shows characteristics independent on incident waves including other than linear polarization such as circular polarization. Therefore, we have revised it to “accommodate both horizontal and vertical polarization (or dual polarization)”.

Fig. R17: Polarization dependency of the transmittance for the S-SRRs. **a** Simulated and **b** measured transmittances, in which the polarization of the incident waves is varied from x-polarization to y-polarization with 15-degree step.

As the reviewer pointed out, the loss tangent of the LC material is large at low frequencies, where the orientation polarization of LC can respond to an external electric field. We discussed the losses at low frequencies with the LC material supplier. For the LC materials commonly used in LCDs, the orientation polarization can respond to an external electric field up to approximately 10 MHz, and the loss becomes small above that frequency [R10]. The LC material used in this study (DHB-012) has a larger molecular weight than those of the LC materials reported in [R10], and it is presumed that the frequency at which the orientation polarization is unable to respond is lower. Therefore, we believe that the loss tangent in the 10 GHz band disclosed by the supplier (see Authors' response #2-7) will not change significantly in the GHz order frequency band. However, as the reviewer noted, the discussion on designable frequencies should be based on empirical evidence, so we have changed the simulated result for 3 GHz to that of 10 GHz where the loss tangent of the LC material has already been measured.

Corresponding revisions

in the revised manuscript: line 100, Figure 2g

in the revised supplemental material: Added a new section (Section 6)

[R10] Kamei, T., Utsumi, Y., Moritake, H., Toda, K. & Suzuki, H. Measurements of the dielectric properties of nematic liquid crystals at 10 kHz to 40 GHz and application to a variable delay line. *Electron. Commun. Japan*, 86, 49–60 (2003).

Comment #2-9

To contextualize novelty, it would be appropriate to discuss recent all dielectric approaches that achieve quasi bound state in the continuum (BIC) resonances with higher Q and negligible ohmic loss, e.g. J. F. Algorri et al., Optics & Laser Technology 161 (2023) 109430, whose silicon slot metasurface offers an instructive comparison for minimizing insertion loss and enhancing modulation depth.

Authors' response #2-9: The authors thank the reviewer for providing interesting information. The employment of BIC resonances in LC metasurfaces has the potential to yield a large modulation depth. Regarding the reported BIC resonances, the fabrication accuracy of the cell structure remains problematic. However, at frequencies around the terahertz band, it should be possible to fabricate them with sufficiently fine accuracy relative to the wavelength. Therefore, if the required dielectric thickness in proportion to the wavelength is not problem, we believe that the use of BIC resonance is a promising approach. Although we have no specific idea yet about the structure, for example, it may be possible to make a cell structure for interaction with incident waves and bias lines and electrodes for controlling the LC orientation without an electrical connection, which may reduce the dissipated power due to bias lines (see Authors' response #2-3). Furthermore, since the ohmic loss of metal patterns tends to increase as the frequency increases, the potential benefits of dielectric-based cells may be further enhanced in the terahertz band. The potential of the using BIC resonances of a dielectric-based cell has been added to the discussion part, along with the relevant references including the proposed one.

Corresponding revision in the revised manuscript:

L345 – L357, Added new references [R11, R12]

[R11] Algorri, J. F. et al. Strongly resonant silicon slot metasurfaces with symmetry-protected bound states in the continuum. Opt. Express 29, 10374–10385 (2021).

[R12] Algorri, J. F. et al. Experimental demonstration of a silicon-slot quasi-bound state in the continuum in near-infrared all-dielectric metasurfaces. Opt. Laser Technol. 161, 109199 (2023).

Comment #2-10

In summary, while the proposed S SRR/LC platform is promising and could indeed influence future transmissive RIS design, the manuscript requires (i) a more rigorous analytical treatment of resonance physics, (ii) quantitative loss budget and thermal analyses, (iii) dynamic response and reliability data, and (iv) expanded bandwidth and polarization measurements before its claims can be considered fully substantiated.

Authors' response #2-10: Again, the authors would like to thank the reviewer for taking the time and energy to review our manuscript. We have worked hard to incorporate your valuable comments and hope that these revisions persuade you to accept our submission.

Reviewer #3:

This paper presents a transmissive metasurface operating in the 115 GHz band that utilizes a thin 3.5 μm liquid crystal (LC) layer for sub-terahertz beamforming. In contrast to conventional approaches that rely on phase control in tunable LC metasurfaces at microwave and millimeter-wave frequencies, the authors propose an amplitude modulation-based control concept.

Strengths:

- *The authors have fabricated a functional sample comprising a large number of elements, demonstrating a scalable approach.*
- *Near-field measurement results are presented.*

Weaknesses:

- *Several technical aspects of the metasurface implementation and performance are not sufficiently explained.*
- *A comparison with state-of-the-art solutions is missing, making it difficult to assess the novelty and competitiveness of the presented approach.*

Authors' response: We thank the reviewer for the time, feedback, and constructive criticism. The comments listed by the reviewer were addressed below with all the diligence and effort to improve the manuscript.

Comment #3-1

Detailed Comments:

1. Missing Performance Metrics:

Several key parameters of the structure are not reported, which prohibits a fair comparison with existing solutions:

- *Response Time: What is the expected or measured response time of the LC device?*
- *Losses: Please provide data on the aperture efficiency of the system.*
- *Bandwidth: A 10% bandwidth is claimed, but the definition used (e.g., -3 dB, -10 dB, ...) is not specified.*

Include a section discussing these performance parameters. In addition, a comparative table listing these values alongside relevant state-of-the-art transmissive or reflective LC metasurfaces must be added to the main manuscript. The structure should also be compared to other thin LC designs available in literature. Other structures similar to these include: reflective metasurfaces, reconfigurable intelligent surfaces (RIS), reflectarrays, transmitarrays, and frequency selective surfaces.

Authors' response #3-1: The authors thank the reviewer for important suggestion. To clarify the novelty of the proposed structure, we have made a table comparing it with other reported transmissive LC metasurfaces. Table R4 compares the proposed unit cell and other published transmissive LC metasurfaces in terms of polarization, LC thickness, wavefront modulation scheme, scanning capability, cell size, and insertion loss.

As mentioned in Authors' response #2-6, the expected response time of the proposed S-SRR was evaluated using a sample with the same LC thickness ($3.5\ \mu\text{m}$) as the proposed structure, exhibiting a rise time of 3 ms and a fall time of 50 ms at a room temperature, 25°C . There are few reports that have evaluated the response time of the transmissive metasurfaces. Therefore, in the table, we compare the LC thicknesses which are strongly correlated with the response time of the LC orientation change. Given that the response time slows down in proportion to the square of the LC thickness [R13], achieving an LC RIS with an LCD level of thickness is important not only for compatibility with the LCD process, but also for achieving a short response time. Besides adding the table, we have updated Fig. S1 in the original supplemental material to Fig. R18 to make it easier to compare the LC thickness with those of other reports, which is the most important parameter in our design. It can be clearly seen that our design has the remarkably thin LC, only $3.5\ \mu\text{m}$, which suggests our design can achieve a relatively fast response.

Regarding losses, it is difficult to fairly evaluate the aperture efficiency of devices such as RISs, where the input power changes depending on the directivity and relative position of the wave source, and there have been few reports evaluating the aperture efficiency of transmissive LC metasurfaces. Although it is difficult to compare aperture efficiencies, the theoretical value of diffraction efficiency, which indicates how much of the power input to the aperture is diffracted in the designed direction, can be estimated by the wavefront modulation scheme of the transmissive metasurface. As described in Authors' response #1-9, the theoretical efficiency of an aperture with amplitude modulation is 10% [R14]. In addition, the efficiency of an aperture with phase modulation can be estimated using the quantization number of the phase range of 2π ; it is 41% for the quantization number of 2 (1 bit) and 81% for the quantization number of 4 (2 bit) [R15]. We believe that including the information regarding the reported wavefront modulation schemes (Amplitude or Phase) and unit cell insertion losses of transmissive LC metasurfaces in the comparative table will facilitate an estimation of the aperture efficiency by readers. Here, two-dimensional beamforming necessitates the use of control bias lines with components parallel to the E field of the incident waves, which affects the interaction between the cell and the waves. Most of the reports have demonstrated only uniform control or one-dimensional beam control. However, when these are extended to two-dimensional control, losses due to bias lines should be considered. In the proposed cell, to achieve two-dimensional beamforming, the control bias lines in two orthogonal directions were connected to each cell, and they were experimentally demonstrated to have a low insertion loss of approximately 2.5 dB for the largest (218×218) array scale. Furthermore, our design has four-fold rotational symmetry including bias lines, which supports both two-dimensional beamforming and dual linear polarization. In relation to Authors' responses #1-9 and #2-4, we also would like to mention that our design has the smallest cell size.

Table R4. Comparison of the proposed cell and other transmissive LC metasurfaces.

Ref.	Frequency [GHz]	Structure	Polarization	LC thickness [μm]	Wavefront modulation	Scanning capability	Cell size (Array size)	Insertion loss
[33] (Meas.)	27.5	Fishnet	Linearly	762 \times 2 layer (0.71 λ \times 2)	Phase (~180°)	1-D	0.42 λ (8 \times 8)	N.A.
[34] (Meas.)	820	SRR	Linearly (Conversion)	800 (2.2 λ)	Amplitude (~30 dB)	Intensity control	0.16 λ (N.A.)	~14 dB
[35] (Meas.)	9.75	Dipole	Linearly	600 (0.02 λ)	Phase (~90°)	1-D	0.36 λ (15 \times 9)	~25 dB
[36] (Sim.)	340	Patch	Linearly	580 \times 2 layer (0.65 λ \times 2)	Phase (~180°)	1-D	0.38 λ (20 \times 20)	2.5 dB
[37] (Meas.)	793	Wires	Linearly	500 (1.32 λ)	Phase (~90°)	N.A.	10.5 λ (N.A.)	N.A.
[38] (Meas.)	690	SRR	Linearly (Conversion)	250 (0.58 λ)	Amplitude (N.A.)	Intensity control	0.23 λ 0.31 λ (150 \times 150)	N.A.
[39] (Meas.)	528	SRR	Linearly	50 (0.088 λ)	Amplitude (~6 dB)	Intensity control	0.1 λ (N.A.)	~19 dB
[40] (Meas.)	345	Patch	Linearly	50 (0.058 λ)	Amplitude (~15 dB)	1-D	0.89 λ (40 \times 40)	~1.2 dB
[41] (Meas.)	421.2	Ring slot	Dual-linearly	45 (0.063 λ)	Amplitude (~30 dB)	Intensity control	0.43 λ (46 \times 46)	1.19 dB
[42] (Meas.)	800	Meandering wires	Dual-linearly	12 (0.032 λ)	Phase (~14°)	2-D	0.22 λ (10 \times 10)	~9.5 dB
Ours (Meas.)	115	SRR based	Dual-linearly	3.5 (0.0013 λ)	Amplitude (12 dB)	2-D	0.12 λ (218 \times 218)	~2.5 dB

Fig. R18: LC-thickness benchmark of the proposed transmissive LC metasurface.

In responses to the reviewer's comment, we have included a discussion about the performance parameters along with the comparative Table 4 below and figure in the main text and removed the original section 1 from the supplemental material. In addition, we have added a note to the main text that the bandwidth is a 3-dB bandwidth

Corresponding revisions

in the revised manuscript: line 276, line 314 – line 344, Added a Table 1 and a Figure 7

in the revised supplemental material: Removed original section 1.

[R13] Jakoby, R., Gaebler, A. & Weickhmann, C. Microwave Liquid Crystal Enabling Technology for Electronically Steerable Antennas in SATCOM and 5G Millimeter-Wave Systems. *Crystals* 10, 514 (2020).

[R14] Oggioni, L., Pariani, G., Zamkotsian, F., Bertarelli, C. & Bianco, A. Holography with Photochromic Diarylethenes. *Materials* 12, 2810 (2019).

[R15] J. W. Goodman. Introduction to Fourier Optics 4th Edition. (W.H. Freeman & Company, 2017).

Comment #3-2-1

2. Amplitude vs. Phase Control:

The choice of amplitude modulation instead of phase control is a central part of this work. This means that the beam steering is greatly suboptimal, since elements not contributing to certain points are not adapted to contribute, but just switched off. Although the approach is interesting, it requires a more detailed explanation:

- The fact that non-contributing elements are merely deactivated leads to suboptimal beamforming performance. This trade-off should be explicitly explained in the main text.

Authors' response #3-2-1: The authors thank the reviewer for these important comments. As you noted, amplitude modulation for wavefront manipulation is an important point in the manuscript. The main reason for employing amplitude modulation is to realize a transmissive-type RIS with a single LC layer. The obtained phase change of waves scattered by a single resonant mode is theoretically limited to π . For reflective metasurfaces, all reflected waves are scattered waves, which means that phase changes due to the metasurface's resonant modes of the metasurface can be fully applied to the reflected waves, and a 2π -range phase control can be achieved by using one designed metasurface layer and ground metal. On the other hand, in the case of a transmissive metasurface, the transmission wave consists of both scattered and non-scattered waves. The non-scattered waves do not undergo phase changes due to resonance modes, which means that a multilayer structure is necessary to achieve controllability over a 2π range. Indeed, Table R4 in

Authors' response #3-1 shows that the phase modulation type employs multiple LC layers. Amplitude modulation can form a 1-bit ($0/\pi$) discrete wavefront with a single LC layer; however, the diffraction efficiency is 10%, which is smaller than the efficiency of phase modulation ($0/\pi$), which is 41%. Thus, a trade-off exists between the metasurface's layer count, i.e., design complexity and fabrication cost, and the efficiency determined by the wavefront modulation scheme. We have included the reasons for employing amplitude modulation and the trade-off relationship to the main text.

Corresponding revisions in the revised manuscript: line 213 – line 221, line 327 – line 337

Comment #3-2-2

- The concept of aperture efficiency for different configurations (e.g., all-on, partially-on) should be quantified and discussed.

Authors' response #3-2-1: Thank you for providing an interesting perspective. We had not considered the case where all-on control is optimal. The amplitude modulation profile changes depending on the relative positioning of the Tx, RIS, and Rx. In most cases, it is necessary to change the direction and focus of the transmitted wave by using amplitude modulation in which on-state and off-state cells coexist in approximately equal proportion, and their theoretical diffraction efficiency is 10%. However, under certain conditions, such as when the Tx, RIS, and Rx are in a straight line, the optimal amplitude modulation profile may be all-on rather than partially-on, as mentioned by the reviewer. When the distance between the Tx and RIS (and between the RIS and Rx) is sufficiently long wherein the difference in path lengths between the transmitting point and the receiving point through each cell of the RIS is less than π , in other words, when the RIS size is smaller than the 1st Fresnel zone, the optimal amplitude modulation is all-on. An aperture with a difference in path lengths through each cell of less than π is equivalent to a 1-bit phase ($0/\pi$) modulated aperture, whose phase quantization error is less than π . Therefore, the diffraction efficiency when all-on is the optimal amplitude modulation profile is 41% or more. Furthermore, when the difference in path lengths between the Tx and Rx via each cell is less than $\pi/2$, the diffraction efficiency of all-on is greater than that for 2-bit phase modulation, i.e., 81% or more, and the diffraction efficiency of all-on gradually approaches 100% as the distance increases. We have included a description of the all-on configuration in the revised manuscript, in addition to the partially-on configuration discussed in Authors' response #1-9.

Corresponding revisions

in the revised manuscript: line 237 – line 241, line 330 – line 337

in the revised supplemental material: Added a new section (Section 8)

Comment #3-2-3

- The reason behind using a binary (on-off) control rather than continuous LC tuning is unclear and needs to be justified, since LC is capable of continuous tuning.

Authors' response #3-2-3: The authors thank the reviewer for pointing this out. As the Authors' response #1-5, in terms of maximizing the received power at the designed position, greyscale control is effective in phase modulation, but there is no benefit to greyscale control in amplitude modulation. We have added the reason for using binary control in this study to the main text.

Corresponding revisions in the revised manuscript: line 296 – line 309

Comment #3-3

3. Control Scheme Explanation:

The authors mention a line-matrix control scheme, but it is unclear why this method is advantageous over an active matrix approach (as in standard LCD displays). Why is pixel-level control via Thin Film Transistors (TFTs) not feasible or practical in this context?

Authors' response #3-3: The authors appreciate your valuable comments on this important matter. Lines 255-261 of the original manuscript describe the possibility of active-matrix control using TFT and its pros (more flexible and ideal wavefront) and cons (increase in the number of control channels, additional loss due to TFT layer) compared with line-matrix control, but there was a risk of giving a negative impression of introducing TFT. Therefore, the description has been revised to emphasize that the introduction of TFT is a promising approach for achieving higher performance LC metasurfaces. Furthermore, an explanation about the advantages of line-matrix control from a cost perspective has been added.

Corresponding revisions in the revised manuscript: line 231 – line 233, line 365 – line 368

Comment #3-4

4. Formal and Presentation Issues:

- The manuscript contains numerous grammatical and stylistic issues. A thorough revision of the English language is recommended.

- The font size in figures and plots is too small and should be increased to match the main text font for readability.

- The introduction has no headline.

- There are several formal issues such as the referencing of figures. Please revise.

Authors' response #3-4: We sincerely thank the reviewer for careful reading and pointing these out. We have checked style and formatting guide of the journal and corrected all the pointed out issues. In addition, a native English speaker and a technical English writing service both reviewed the English grammar of the revised manuscript and supplemental material again and the incorrect parts were revised.

Comment #3-5

Taking these comments into account, I believe the authors should revise these comments before the manuscript can be accepted.

Authors' response #3-5: The authors would like to thank the reviewer again for taking the time and energy to review our manuscript. The reviewer's insightful feedback has improved our manuscript's quality and made its novelty clear. We hope these revisions persuade you to accept our submission.

Reviewer #4:

“I co-reviewed this manuscript with one of the reviewers who provided the listed reports. This is part of the Communications Engineering initiative to facilitate training in peer review and to provide appropriate recognition for Early Career Researchers who co-review manuscripts.”

Authors' response: The authors sincerely appreciate the reviewer's effort on our manuscript. All the concerns from the reviewers are addressed in detail in the point-to-point response.

Response to Reviewers (2nd round)

Manuscript ID: COMMS-ENG-25-0201-T

“Transmissive metasurface with 3.5- μm -thick liquid crystals for subterahertz-wave dynamic beamforming”

We thank the reviewers for their continued effort and constructive feedback. Below we provide point-by-point responses to the second-round comments, with the authors’ responses in blue. We have also included a revised version of the manuscript and supplemental material with changes in red.

Reviewer #1:

The authors explain the concerns, providing detailed reasoning, labeling the corresponding revisions, and adding supporting references. The improved manuscript quality enables readers to understand the technical details and underlying mechanisms more clearly. The work being reported is of good quality, with essential materials displayed to demonstrate its functionalities. I recommend publication as is.:

Authors’ response: We sincerely thank the reviewer for the positive assessment and recommendation for publication. The reviewer’s valuable feedback was crucial for us to improve the manuscript. Again, we would like to express our gratitude for your high expertise and insightful feedback.

Reviewer #2:

Summary, scope, and principal claims

The paper reports a transmissive reconfigurable intelligent surface (RIS) at 115 GHz based on a single-layer, 3.5 μm liquid-crystal (LC) gap and a stacked split-ring resonator (S-SRR) unit cell operated in a circulating-current mode. The demonstrator comprises 218 \times 218 cells in a 70 mm aperture (cell pitch $< \lambda/8$) addressed by line-matrix control with 109 row + 109 column channels, enabling binary amplitude modulation for 2-D collimation and beam steering from 0° to 30° in 5° steps. The authors measure an on/off modulation depth ≈ 12 dB, near-field maps, and derive far-field patterns from a channel model; they provide design files on request. Key device and method details are consolidated in the revised manuscript and Supplementary Information.

Novelty and relation to prior work

Prior LC-based THz metasurfaces and RISs have typically used hundreds-of-micrometres LC thickness, often in reflective mode or with 1-D steering. A representative transmissive 1-bit LC coding metasurface (AOM 2021) showed programmable THz beam manipulation but with much thicker LC and smaller arrays. In parallel, a Communications Engineering 2024 study achieved sub-100 ms LC-RIS response with compact delay lines, but at 62 GHz and in reflection, not transmission; it is a useful architectural comparator. Related THz programmable metasurface work has explored crossbar addressing and 2-D steering, mainly reflectively. An Optics Letters 2022 paper also discusses LC-programmable THz metasurfaces. Against this backdrop, the LCD-grade 3.5 μm LC gap at 115 GHz in transmission and the large-aperture, 2-D line-matrix control are the most distinctive aspects here. The manuscript's comparison table and LC-thickness benchmark underscore this positioning. Overall, the work should interest both metasurface and 6G/RIS communities.

Technical soundness

Unit-cell physics and materials. The rebuttal now gives a clearer physical rationale for abandoning the anti-symmetric (magnetic-dipole) bilayer resonance at thin gaps by quantifying Q -factor degradation under realistic metal conductivity and LC loss tangent, and it introduces a hybridization/equivalent-circuit view for the S-SRR. These additions materially improve rigor.

Bias lines and loss budget. The revised analysis identifies ITO bias lines ($60 \Omega/\square$) as the dominant non-ideal contributor at 115 GHz ($\sim 20\%$ dissipated power at resonance; transmittance peak limited to ~ 17 dB with ITO), correctly highlighting that raising line impedance is the most effective path to higher modulation depth. The main text and Supplementary quantify power dissipation pathways and explicitly note the approximation in the channel model. Aperture formation and steering. The beam-forming strategy (binary amplitude patterns P0–P7) is coherently tied to the Fresnel-zone condition (all-on optimal when the RIS is smaller than the first Fresnel zone) and to the known 10% theoretical efficiency ceiling of amplitude-only holography; these points are stated and referenced. The measured near-field maps versus calculated reference fields are consistent with the intended steering behaviour across 0° – 30° , for both orthogonal linear polarisations.

Bandwidth and polarisation. The authors appropriately temper “broadband/polarization-independent” claims to $\sim 10\%$ fractional bandwidth around 115 GHz and dual linear polarisation, and they add simulations/measurements sweeping the incident polarisation from 0° to 90° .

Dynamics and stability. Because the 218-ch feasibility driver is not speed-optimised, the authors characterise LC orientation response optically on a 3.5 μm cell (≈ 3 ms rise at 25°C ; ≈ 50 ms fall), and discuss UV/thermal stability and mitigation (UV blocking, thermal management). These additions address prior concerns, albeit indirectly with respect to THz modulation.

Adequacy of responses to reviewers

The point-by-point rebuttal is careful and largely convincing:

Resonance physics and linewidth/Q—addressed with new simulations, geometry tables, and an equivalent-circuit discussion. OK

Loss budget & thermal stability—quantified bias-line loss; qualitative but reasonable thermal/UV discussion. Partially. (thermal data still limited).

Near- to far-field validity—channel model derived (Eq. S14) with explicit caveats (no polarisation/loss). Partially; a direct far-field validation remains desirable.

Bandwidth/polarisation claims—language corrected; additional polarisation sweeps provided.

Driver and reproducibility—schematic of the switch matrix (MOSFET relays) added; layout files available; fabrication alignment sensitivity and a two-sample comparison discussed.

Overall, the authors have substantially strengthened the manuscript; the remaining gaps are mainly experimental breadth (see below).

Authors' response: We greatly appreciate the time and expertise the reviewer invested in reviewing both the corrected paper and our previous point-by-point responses. The reviewer's constructive comments and suggestions were invaluable in further improving the technical clarity and quality of our work. In response to the current round of feedback, we have conducted additional experiments and simulations wherever feasible and engaged in further discussions with the LC material supplier to better address the remaining concerns. Below, we provide detailed point-by-point responses to each comment.

Comment #1:

Statistical analysis and reproducibility

The work is primarily demonstration-driven; statistical treatment is minimal. There are no error bars on S-parameters or near-field magnitudes, no repeated-sample statistics beyond the alignment illustration, and no quantified uncertainties from probe positioning or calibration. The channel model is clearly stated, but its simplifications are acknowledged.

Reproducibility improved with complete geometry (Supp. Sec. 12), driver schematic, and data/layout availability, which is commendable.

Recommendation: include uncertainty estimates (repeat scans; calibration repeatability; sensitivity to V_p) and, if feasible, a single far-field measurement on a scaled-down aperture or with a compact range to validate the model-based far-field claims.

Authors' response: We sincerely thank the reviewer for the comments regarding statistical treatment and reproducibility. We fully agree that statistical analysis is essential in wireless device evaluation, especially because propagation environments are always changing and can greatly affect the results of measurements. In the manuscript, all measurements were conducted in an anechoic chamber, ensuring a static and controlled propagation environment. Nevertheless, to estimate measurement uncertainty, we performed repeated measurements (Fig. R2-1) based on the

far-field evaluation described in the following Authors' response. These repetitions confirmed that the measurement uncertainty, including the influence of unintended reflections between instruments, remains below ± 0.2 dB. We have added this uncertainty estimate to the measurement section in the revised manuscript.

Fig. R2-1: (a) Repeated far-field measurement results for the collimating pattern with reduced aperture at 115 GHz. (b) Variation in received power across repeated scans, confirming measurement uncertainty remains below ± 0.2 dB.

Regarding probe positioning sensitivity, the evaluation was carried out through the focusing characteristics, which represent a highly sensitive condition. Figure R2-2 shows the measured position dependence of the Tx probe antenna on the received power at the focal point when the Tx antenna is positioned at 100 mm, and the focal length of the transmissive waves is 50 mm from the LC metasurface.

We found that the sensitivity in the horizontal direction (parallel to the metasurface plane) is higher than in the vertical direction (normal to the metasurface plane). Nevertheless, even in the horizontal direction, the variation in received power remains within approximately 1 dB when the probe position is maintained within ± 2 mm from the design position. Given that the positioning system used (SGSP46, SIGMAKOKI Co., Ltd.) offers a positioning accuracy of $6 \mu\text{m}$, we believe the reproducibility of probe positioning is sufficiently ensured.

Fig. R2-2: Received power variation at the focal point on the probe displacement (a) in the horizontal direction (parallel to the metasurface plane) and (b) in the vertical direction (normal to the metasurface plane).

As for the sensitivity to the applied voltage (V_p), the measured modulation depth varies significantly across the V_p range of approximately 1 V to 3 V, with a maximum slope of approximately 7 dB/V (Fig. R2-3). In our experiments, however, we used $V_p = 0$ V for the off-state and $V_p = 8$ V for the on-state, where the modulation depth saturates and becomes insensitive to V_p changes. Therefore, we believe the reproducibility of our results is not influenced by V_p sensitivity.

Fig. R2-3: Measured modulation depth dependency on applied voltage V_p .

Corresponding revisions

in the revised manuscript: line 209 – line 210, line 439 – line 440, line 454 – line 457

in the revised supplemental material: Added a new section (Section 16)

Comment #2:

Literature coverage

The revised text now references dielectric BIC metasurfaces as a promising low-loss path, aligning with reviewers' suggestions. Two important, still-missing comparators are recommended for citation: (i) transmissive LC 1-bit coding metasurface at THz frequencies (AOM 2021), which contextualises transmissive, digitally addressed LC devices, and (ii) Comms Eng. 2024 sub-100 ms LC-RIS with compact delay lines, which is reflective and lower-frequency but directly relevant to the system-level control architecture and response-time targets. For completeness on crossbar/2-D addressing at THz, citing recent Science Advances 2023 on modulo-addition steering would strengthen the discussion of discrete optimisation and line-matrix trade-offs.

Authors' response: We sincerely thank the reviewer for the valuable suggestions regarding relevant literature. As pointed out, the three recommended works are closely related to our study and provide important context for transmissive LC metasurfaces, system-level control architectures, or discrete optimization strategies.

In response, we have added citations to the following papers in the revised manuscript:

[R2-1] Neuder, R., Späth, M., Schüßler, M. & Jiménez-Sáez, A. Architecture for sub-100 ms liquid crystal reconfigurable intelligent surface based on defected delay lines. *Commun. Eng.* **3**, (2024).

[R2-2] Liu, C. X. *et al.* Programmable Manipulations of Terahertz Beams by Transmissive Digital Coding Metasurfaces Based on Liquid Crystals. *Adv. Opt. Mater.* **9**, 2100932 (2021).

[R2-3] Li, W. *et al.* Modulo-addition operation enables terahertz programmable metasurface for high-resolution two-dimensional beam steering. *Sci. Adv.* **9**, (2023).

We have also expanded the discussion in the manuscript to reflect the relevance of these works, particularly in the context of control architecture. We appreciate the reviewer's guidance in helping us strengthen the positioning and completeness of our literature review.

Corresponding revisions in the revised manuscript:

line 72 – line 76, line 233 – line 236, Added new references [R2-1–R2-3], Table 1, Figure 7

Comment #3:

Weaknesses and suggested experiments/analyses

Dynamic THz response. The optical ms-scale response is informative but should be complemented by THz-band electrical switching (e.g., rise/fall of the on/off ratio at 115 GHz under the revised driver).

Authors' response: We sincerely thank the reviewer for highlighting the importance of complementing the optical response measurements with direct THz-band switching characterization. We agree that measuring the rise and fall times of the switching behavior at 115 GHz would provide the most direct validation. However, performing THz-band switching time measurements with our current proof-of-concept driver and measurement setup is challenging. As an alternative, we have discussed with the LC material supplier (DIC Corp., Japan) and simulated the dynamics of the LC molecule (director) within the proposed metasurface structure at 25°C. This simulation (LCD Master, Shintech Optics) analyzed the dynamics of the director based on device geometry and LC material properties. We input our metasurface structure and the material parameters of the used LC material (DHB-012) into the model. The simulation results, shown in Fig. R2-4, reveal that when V_p varies from 0 to 8 V, the director tilt reaches approximately 90° within 3 ms and subsequently stabilizes. When V_p returns from 8 to 0 V, the maximum tilt decreases below 30° within 50 ms and approaches a nearly horizontal alignment (<10°) by 100 ms.

Fig. R2-4: Simulation results of LC molecule dynamics at 25°C in the S-SRR. (a) S-SRR model used in the simulation of LC director dynamics. (b) Director tilt distribution in the cross-section of the LC layer of the S-SRR for the rise time and (c) the fall time evaluation.

These simulated results align with the measured optical ms-scale response and support the advantage of using an LCD-grade LC thickness.

In response, we have added these results and discussion to the revised supplemental material. We believe this analysis provides meaningful insight into the dynamic behavior of the LC metasurface, even though direct THz-band switching measurements remain a future goal.

Corresponding revisions in the revised supplemental material:

line 377 – line 385, Added new figure (Fig. S16)

Comment #4:

Far-field validation. Provide at least one direct far-field data set (reduced aperture or compact-range) to benchmark the S14-based conversion.

Authors' response: We thank the reviewer for the excellent suggestion to validate the channel-model-based calculations with a direct far-field measurement with a reduced aperture. Following this suggestion, we performed an additional experiment in which the effective aperture of the LC metasurface was reduced to 30-mm square using absorbers (Fig. R2-5). The far-field condition can be expressed as

$$R_{\text{far}} > \frac{2D^2}{\lambda}, \quad (R10)$$

where D is the diameter of the aperture and λ is the wavelength. For $D = 30$ mm and $\lambda = 2.6$ mm (115 GHz), the far-field distance, R_{far} , is approximately 0.69 m. To ensure a sufficient margin, we placed the measurement plane at 1 m from the metasurface plane. Figure 2-6a compares the measured electric-field distribution in the xy -plane at $z = 1$ m for the all-on state and the collimating control state with the calculated distribution using the channel model, showing good agreement between the measured and calculated results. Figure R2-6b shows the electric-field profiles along $y=0$ mm, indicating that the gain improvement due to collimation is approximately 6 dB—limited by the reduced aperture size—but the measured and calculated gains match closely. We believe that these results validate the operation of the fabricated LC metasurface and the channel-model-based calculations under far-field conditions.

We have added these results and discussion to the revised supplemental material.

Fig. R2-5: Experimental setup for far-field validation using a reduced aperture (30 mm × 30 mm).

Fig. R2-6: Comparison of measured and simulated electric-field profiles at $z = 1$ m for the reduced aperture. (a) Measured and calculated electric-field distribution on the xy plane at $z = 1$ m for all-on and collimating control. (b) Measured and calculated relative gain along $y = 0$ mm, showing ~ 6 dB gain due to collimation and good agreement between measurement and calculation.

Corresponding revisions

in the revised manuscript: line 192 – line 194

in the revised supplemental material: Added a new section (Section 6)

Comment #5:

Loss definition consistency. The claim of low “insertion loss” at the array scale should be reconciled with per-cell S_{21} (≈ -10 to -14 dB) to avoid ambiguity between transmission magnitude and holographic aperture efficiency. Clarify the metric used in the comparison table.

Authors’ response: Thank you for pointing this out. We agree that the definition of insertion loss was not sufficiently clear in the original manuscript. In the comparison table, the insertion loss corresponds to experimentally measured transmission losses for a uniformly controlled cell array. For amplitude-modulation metasurfaces, the insertion loss refers to the transmission loss at the operating frequency when the cell is in the state with the highest transmittance (the “on-state” in our manuscript). For phase-modulation metasurfaces, the insertion loss is the value at the operating frequency in the state that exhibits the lowest loss among the available phase states.

To avoid ambiguity, we have added an explanation of this definition in the revised manuscript.

Corresponding revisions in the revised manuscript: line 320 – line 325

Comment #6:

Polarisation and oblique incidence. The new polarisation sweep is useful; adding oblique-incidence data (even simulated) would frame window-mounted deployments more realistically.

Authors’ response: We are grateful for this valuable suggestion. We agree that oblique-incidence characteristics are highly relevant for practical scenarios such as window-mounted deployment of LC metasurfaces. In accordance with the suggestion, we have performed additional full-wave simulations to analyze S-SRR’s performance under oblique incidence from 0° to 60° for both TE- and TM-polarized incident waves (Fig. R2-7). For TE-polarized incidence, an increase in the incidence angle results in higher insertion loss in both on-state and off-state, and the resonance peak becomes deeper. On the other hand, for TM-polarized incidence, increasing the angle results in lower insertion loss in both states, accompanied by a shallower resonance peak. The modulation depth at 115 GHz exhibits opposite trends for TE- and TM-polarized incidence. The decrease in modulation depth for TM-polarized incidence was approximately 2 dB even at 60° , indicating that the proposed structure maintains effective performance over a wide range of incidence angles.

Fig. R2-7: (a) Simulated transmission spectra for TE-polarized incidence and (b) TM-polarized incidence at angles from 0° to 60° . (c) Modulation depth on the incidence angle for TE- and TM-polarized incidence at 115 GHz.

Corresponding revisions

in the revised manuscript: line 206 – line 208

in the revised supplemental material: line 245 – line 253, Added a new figure (Fig. S10)

Comment #7:

Reliability. Consider including accelerated UV/thermal stress results or supplier specifications for DHB-012 to substantiate the stability discussion.

Authors' response: Thank you very much for this important comment. We agree that long-term reliability under UV and thermal stress is a critical issue for practical deployment. While we requested accelerated UV and thermal stress data for DHB-012, the supplier was unable to provide this information because it is considered confidential. As an alternative, we have expanded the discussion by citing relevant literature that highlights UV and heat as major factors affecting the stability of liquid crystal. For example, previous studies have shown that LC materials exhibit absorption in the UV band, making them susceptible to UV-induced degradation [R2-4]. In addition, a comprehensive review of failure mechanisms in LCDs under environmental stress emphasizes that both UV and heat significantly degrade LC materials and that mitigation strategies are essential for outdoor applications [R2-5].

Based on these findings, we emphasize that UV mitigation strategies, such as UV-cut glass, UV-blocking films, and protective coatings, are indispensable for practical implementation in the revised manuscript. These approaches are well-established in the LCD industry and are equally relevant for metasurface-based devices. Furthermore, future work should include accelerated UV and thermal stress testing to quantitatively evaluate long-term stability and identify potential

failure mechanisms under realistic outdoor conditions. We have added this point to the revised manuscript to highlight the importance of reliability assessment for practical deployment.

[R2-4] Lin, P. T., Wu, S. T., Chang, C. Y. & Hsu, C. S. UV Stability of High Birefringence Liquid Crystals. *Mol. Cryst. Liq. Cryst.* 411, 243–253 (2004).

[R2-5] Gue, R. Understanding Heat-Induced Liquid Crystal Display Failures. SSRN. doi:10.2139/SSRN.5216972 (2024).

Corresponding revisions in the revised manuscript:

line 394 – line 406, Added new references [R2-4–R2-5]

Comment #8:

Editorial/minor

The authors addressed numerous clarity issues (terminology, scale bars, bandwidth phrasing), and the Data availability statement now explicitly mentions DXF layout provision—good.

Technically convincing with important caveats. The manuscript demonstrates a credible path to LCD-compatible, thin-gap LC transmissive RIS at 115 GHz with 2-D beam control and provides significantly improved physical and methodological grounding after revision. Addressing the THz-band switching dynamics, a minimal far-field validation, and metric consistency for “insertion loss” would remove the remaining doubts. I recommend acceptance after minor–moderate revision, contingent on clarifications/experiments as outlined above.

Authors’ response: Again, the authors greatly appreciate the reviewer’s insightful comments. In this response, we tried to address all the comments from the reviewer as thoroughly as possible. We believe these revisions address the outstanding concerns.

Reviewer #3:

Thank you for the thorough revision of the paper. We believe that our previous comments have been well addressed and that the technical quality of the manuscript has improved substantially.

Before accepting the paper, however, we find that the abstract remains superficial and should be improved to give the reader a clearer impression of the paper's content. In particular, the following aspects should be mentioned more explicitly:

1 - The use of amplitude modulation

2 - The key performance metrics, i.e., bandwidth, insertion loss, scanning capabilities, cell size, and polarization. Please provide quantitative data of these metrics already in the abstract.

Authors' response: We sincerely appreciate the reviewer's expertise and constructive feedback. We have revised the abstract to explicitly state that our LC metasurface employs amplitude modulation and incorporated key performance metrics. In addition, the abstract was shortened for clarity and to follow the journal's style and formatting guide.

Corresponding revisions in the revised manuscript: line 13 – line 25

Reviewer #4:

"I co-reviewed this manuscript with one of the reviewers who provided the listed reports. This is part of the Communications Engineering initiative to facilitate training in peer review and to provide appropriate recognition for Early Career Researchers who co-review manuscripts."

Authors' response: The authors sincerely appreciate the reviewer's hard work and valuable contributions to our manuscript during the review process.

Reviewer #5:

"I co-reviewed this manuscript with one of the reviewers who provided the listed reports. This is part of the Communications Engineering initiative to facilitate training in peer review and to provide appropriate recognition for Early Career Researchers who co-review manuscripts."

Authors' response: The authors sincerely appreciate the reviewer's hard work and valuable contributions to our manuscript during the review process.

Response to Reviewers (3rd round)

Manuscript ID: COMMS-ENG-25-0201-B

“Transmissive metasurface with 3.5- μm -thick liquid crystals for subterahertz-wave dynamic beamforming”

We thank the reviewers for their continued effort and constructive feedback. Below we provide point-by-point responses to the third-round comments, with the authors' responses in blue.

Reviewer #2:

All my comments were properly addressed. I suggest acceptance in the current form.

Authors' response: We sincerely thank the reviewer for the positive assessment and recommendation for publication. The reviewer's valuable feedback was crucial for us to improve the manuscript. Again, we would like to express our gratitude for your high expertise and insightful feedback.